# Bladder cancer variants share aggressive features including a CA125+ cell state and targetable TM4SF1 expression

Heiko Yang[1,2,11], Hanbing Song [3,11], Elizabeth Yip[3], Timothy Gilpatrick[4], Kevin Chang[1,3], Paul Allegakoen[3], Kevin L. Lu [4], Keliana Hui[3], Julia H. Pham[3], Corynn Kasap[3], Vipul Kumar[3], Janae Gayle[3,5], Bradley A. Stohr[4], Chien-Kuang Cornelia Ding [4], Arun P. Wiita [6], Maxwell V. Meng[1], Jonathan Chou [3], Sima P. Porten[1,12] & Franklin W. Huang [1,3,7,8,9,10,12] ✉

Histologic variant (HV) subtypes of bladder cancer are clinically aggressive tumors that are more resistant to standard therapy compared to conventional urothelial carcinoma (UC). Little is known about the transcriptional programs that account for their biological differences. Here we show using single cell analysis that HVs harbor a tumor cell state characterized by expression of *MUC16* (CA125), *MUC4*, and *KRT24*. This cell state is enriched in metastases, predicted to be highly resistant to chemotherapy, and linked with poor survival. We also find enriched expression of *TM4SF1*, a transmembrane protein, in HV tumor cells. Chimeric antigen receptor (CAR) T cells engineered against TM4SF1 protein demonstrated in vitro and in vivo activity against bladder cancer cell lines in a *TM4SF1* expression-dependent manner, highlighting its potential as a therapeutic target.

Histologic variant (HV) subtypes of bladder cancer are found in up to 25% of all bladder tumors. Compared to bladder tumors with pure urothelial carcinoma (UC) histology, tumors with HVs are associated with worse clinical outcomes[1,2]. The optimal clinical management of HV bladder cancers remains challenging as many HV subtypes do not respond well to systemic therapy and the need for better treatment options represents a major unmet need[3–7].

While significant progress has been made to define the molecular characteristics of pure UC[8–10], much less is known about the biology of HVs. Fundamentally, it remains unclear whether each HV subtype should be regarded as a distinct entity or whether HVs share common features as a group. Some genomic alterations, such as *TERT* promoter mutations in micropapillary, plasmacytoid, and adenocarcinoma variants, appear to be more associated with HVs than UCs, while others, such as *CDH1* truncations in plasmacytoid variants, are thought to be subtype defining[11–15]. While Eemerging evidence suggests that HV biology may not be governed solely at the genomic level, transcriptional analyses based on bulk RNA sequencing data remain limited to date[16,17]. The use of bulk RNA sequencing is not well suited to study HVs because it requires large sample sizes that are difficult to achieve in HVs, especially when considering heterogeneity related to number of individual subtypes.

In this work, we generated a single-cell RNA sequencing (scRNA-seq) atlas of HV bladder tumors to gain further insight into the

[1]Department of Urology, University of California San Francisco, San Francisco, CA, USA. [2]Division of Urology, Department of Surgery, University of Colorado, Aurora, CO, USA. [3]Division of Hematology/Oncology, Department of Medicine, University of California San Francisco, San Francisco, CA, USA. [4]Department of Pathology, University of California San Francisco, San Francisco, CA, USA. [5]College of Letters and Science, University of California Santa Barbara, Santa Barbara, CA, USA. [6]Department of Laboratory Medicine, University of California San Francisco, San Francisco, USA. [7]Chan Zuckerberg Biohub, San Francisco, CA, USA. [8]Bakar Computational Health Sciences Institute, University of California San Francisco, San Francisco, CA, USA. [9]Institute of Human Genetics, University of California San Francisco, San Francisco, CA, USA. [10]Division of Hematology and Oncology, Department of Medicine, San Francisco Veterans Affairs Medical Center, San Francisco, CA, USA. [11]These authors contributed equally: Heiko Yang, Hanbing Song. [12]These authors jointly supervised this work: Sima P. Porten, Franklin W. Huang. ✉e-mail: franklin.huang@ucsf.edu

aggressive biology of HVs and to identify potentially targetable molecular features. We find that HVs harbor a CA125+ cancer cell state that is associated with metastasis and resistance to chemotherapy. We also find enriched expression of *TM4SF1*, a transmembrane protein that can be targeted with CAR T cells in a mouse xenograft model. Our results highlight the potential for scRNA-seq to advance precision cancer medicine approaches in rare, understudied tumors.

## Results

### Single cell analysis of tumor epithelial cells reveals a CA125+ tumor cell state in histologic variants

We collected tissue and dissociated single cells from 4 pure urothelial carcinomas (UC) and 11 histologic variant (HV) tumors. Detailed clinical information is displayed in Supplemental Table 1; pathologic diagnoses were confirmed in specimens collected for sequencing (Supplementary Fig. 1). Single-cell RNA sequencing (scRNA-seq) was performed using the Seq-Well platform, and the sequencing results were processed in our customized analytical pipeline (Supplementary Fig. 2A). After ambient RNA decontamination and removal of low-quality cells, 21,533 cells in total were captured for downstream analysis from these specimens (Supplementary Fig. 2B). While tumor epithelial cells were captured from almost all tumors, the capture rate for stromal and immune cells was highly variable among the specimens (Supplementary Fig. 2C) per our annotation based on graphical clustering patterns and canonical cell-type specific markers for tumor epithelial/urothelial cells (*EPCAM*, *KRT7*), immune cells (*PTPRC*), stromal cells (*DCN*, *ACTA2*), and endothelial cells (*SELE*) (Supplementary Fig. 3A).

We focused our analysis on tumor cell biology by subsetting and re-clustering the tumor epithelial cells from the main dataset (Fig. 1A). We excluded three tumors that did not meet a threshold of 150 tumor epithelial cells for analysis (UC04, VAR10, VAR11). Although neuroendocrine tumors are generally considered non-urothelial cancers[2], we included the tumor with small cell HV (VAR09) due to the presence of urothelial components within the tumor (carcinoma in situ and micropapillary variant). The final tumor epithelial dataset thus included three pure UCs (UC01-UC03) and nine HVs (VAR01-VAR09). To confirm the tumor content in this dataset, we used InferCNV to estimate the copy number profiles of all epithelial cell clusters using stromal and immune cells as reference (Supplementary Fig. 3B). Of note, we did not discern any patterns in copy number variation between HVs vs UCs.

Most tumor cells formed their own clusters corresponding to the tumor of origin and were named accordingly, i.e., VAR01c was the predominant cluster obtained from the VAR01 tumor (Fig. 1A). Although there were few apparent transcriptional similarities among the three micropapillary tumors or between the two nested tumors, we did find an enrichment of genes related to plasma cell maturation[18–29] and small cell lung cancer[30–32] within the top differentially expressed genes (DEGs) in VAR08c (derived from plasmacytoid HV) and VAR09c (derived from small cell HV), respectively (Supplementary Fig. 4A–D). These data indicate that HVs can adopt the transcriptional programs of similar-appearing non-urothelial cells.

One cluster, which we named "Cluster 13" based on the number assigned by the clustering algorithm, was comprised of cells from multiple HV tumors (Fig. 1B, C). This cluster was present in our full dataset with and without integration (Supplementary Fig. 5). Differentially expressed genes (DEGs) for each tumor cluster were computed and curated, and *MUC4*, *MUC17*, *MUC16*, *KRT24,* and *WISP2* were among the top DEGs for Cluster 13 (Fig. 1D). To validate the existence of Cluster 13 cells histologically, we performed immunostaining of MUC4, CA125 (encoded by *MUC16*), and KRT24. We observed close colocalization of MUC4 and CA125 in a subpopulation of tumor cells and a stronger KRT24 signal corresponding to these cells in two HVs (Fig. 1E). We chose CA125 as a surrogate marker for Cluster 13 in a larger bladder

cancer cohort (14 HV tumors, 20 UC tumors) due to its extensive history as a tumor marker in other cancers and the availability of clinically validated antibodies.

In this cohort, we found a subpopulation of CA125+ cells in a variety of HV tumors with different subtypes (13/14) (Fig. 1F) but rarely in tumors with high-grade UC (1/11) or carcinoma in situ (CIS) histology (1/9). In tumors with mixed HV and UC components such as VAR03 and VAR05, CA125+ cells were present in the HV regions (Fig. 1F, pleomorphic giant cell-like, nested) but absent in the high-grade UC and CIS regions (Supplementary Fig. 6). We also did not detect the Cluster 13 signature or expression of *MUC16*, *KRT24*, and *WISP2* in a previously published bladder cancer scRNA-seq dataset derived from UC bladder tumors (Chen et al., Supplementary Fig. 7A, B)[33]. Our results suggest that the cancer cells found in Cluster 13 represent a tumor cell state highly specific to, but not exclusive to, HV-containing tumors. To explore whether CA125 expression in these cells could be useful as a clinical biomarker, we prospectively assayed preoperative serum CA125 levels in bladder cancer patients undergoing surgery and found CA125 levels to be higher in those with HV tumor components in their final pathology compared to those with UC only ($22.7 \pm 6.6$ U/mL vs $11.6 \pm 8.8$ U/mL, $p = 0.007$) (Fig. 1G).

### Cluster 13 cells exhibit hallmarks of transcriptional convergence

To investigate the overall transcriptomic relationship among tumor clusters and to test whether similar HV subtypes share gene expression programs (e.g., micropapillary to micropapillary, nested to nested), an unsupervised partition-based graphical abstraction (PAGA) graph was generated. While we did not observe any prominent subtype-specific associations, we found that Cluster 13 cells formed a central node with an association to almost every other tumor cluster, even to those whose parent tumor did not contribute any cells to Cluster 13 (Fig. 2A). This result raised the possibility that Cluster 13 represents either a convergent cell state or a common progenitor cell state shared by various HV subtypes.

We thus sought to infer the relationship between the Cluster 13 cells and the parent tumor cells. For the five HV tumors that had the highest Cluster 13 content (VAR01, VAR03, VAR05, VAR06, and VAR07), we found that Cluster 13 cells bore the signature of the parent tumor with a high degree of specificity, supporting the likelihood that all cells within each tumor are clonally related (Fig. 2B). On pseudotime analysis, the Cluster 13 cells were selected as the starting point for the pseudotime trajectory in each tumor (Fig. 2C). The Cluster 13 signature was anti-correlated with the parent tumor signature in four of five tumors (Fig. 2D), and the marked dichotomy of the Cluster 13 signature along the pseudotime in all five tumors suggests that Cluster 13 arises as a derivative of the parent tumor rather than vice versa. To further exclude the possibility that Cluster 13 is a progenitor cell state rather than a derivative tumor cell state, we generated a nine-gene bladder stem cell signature (*PROM1* (CD133), *POU5F1* (Oct4), *SOX2*, *ALDH1A1*, *SOX4*, *EZH2*, *YAP1*, *CD44*, and *KRT14*) based on previous studies in bladder cancer stem cells[34–36]; we found no significant enrichment of this signature in Cluster 13 cells (Supplementary Fig. 8). While scRNA-seq alone cannot prove the temporal relationship between these cells, our results support the idea that cancer cells found in Cluster 13 are a convergent cell state that is more prevalent in HV tumors.

### Cluster 13 cells harbor adverse molecular features

Gene ontology (GO) analysis of the Cluster 13 signature revealed a significant enrichment in epithelial-to-mesenchymal transition (EMT) and *KRAS* signaling gene sets (Fig. 3A) suggesting that Cluster 13 cells have more aggressive metastatic potential compared to non-Cluster 13 cells within the same tumor. Using CA125 again as a putative marker for Cluster 13 cells, we examined CA125 staining in five HV tumors with lymph node metastases and observed a striking homogeneous

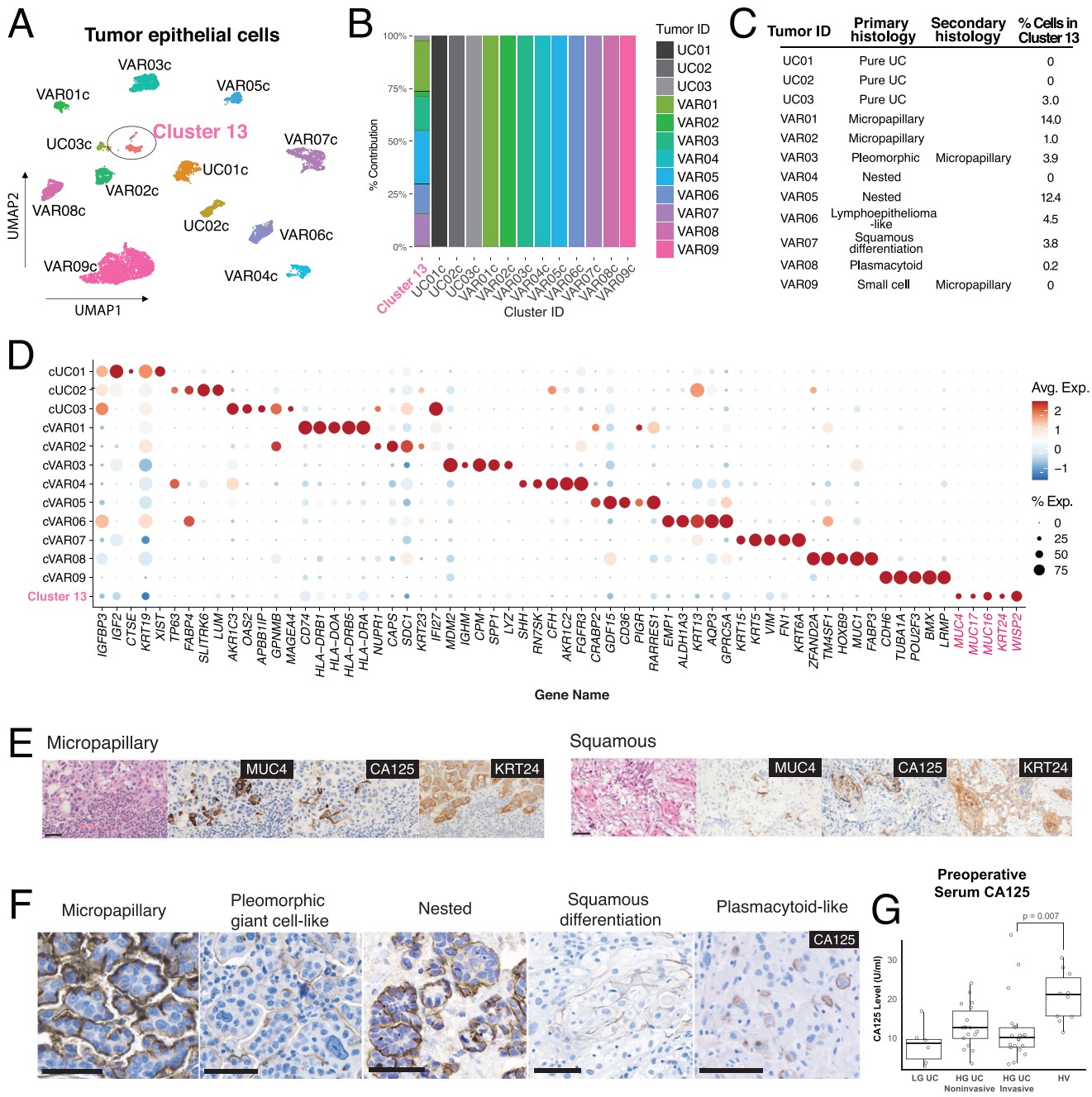

**Fig. 1 | Top level clustering analysis of tumor epithelial cells and characterization of a common tumor cluster. A** Clustering UMAPs of tumor epithelial cells ($N = 8553$) extracted from the main dataset color-coded by cluster and annotated according to tumor ID. Cluster 13 (ellipse) is annotated separately due to contributions from multiple tumors. **B** Cluster composition by patient/tumor. **C** Table displaying primary and secondary histologic patterns observed in each tumor and percentage of cells within Cluster 13. **D** Curated dot plot of top differentially expressed genes (DEGs) by tumor cluster. **E** Serial section immunohistochemistry of MUC4, CA125, and KRT24 in variant tumors ($N = 2$ of 2 cases). **F** Representative immunohistochemistry of CA125 in multiple histologic variants. All scale bars = 50 μm. **G** Preoperative serum CA125 values in bladder cancer patients stratified by tumor grade and histology (Two-way Mann–Whitney U-test, $p = 0.007$). Quartiles, centers and outliers are shown in the box plot. Source data are provided as a Source data file.

enrichment of CA125+ cells in the lymph nodes compared to the primary tumor in 4 of 5 cases (Fig. 3B).

We next evaluated the susceptibility of Cluster 13 cells to chemotherapy and targeted agents in silico. By training a drug response model using the Cancer Drug Response prediction using a Recommender System (CaDRReS) based on the Cancer Cell Line Encyclopedia (CCLE) database and Genomics of Drug Sensitivity in Cancer (GDSC2) database, the estimated efficiency (percentage of tumor cells killed) for drugs from the GDSC2 database was inferred for each tumor cluster in our scRNA-seq dataset (Supplementary Fig. 9)[37]. We performed a side-by-side analysis of Cluster 13 and parent tumor cells within VAR01, VAR03, VAR05, VAR06, and VAR07: in 4 of 5 cases, the Cluster 13 subset was more resistant to conventional bladder cancer agents such as cisplatin, gemcitabine, methotrexate, vinblastine, doxorubicin, and mitomycin C compared to their respective parent tumors (Fig. 3C). Consistent with these adverse features, tumors in

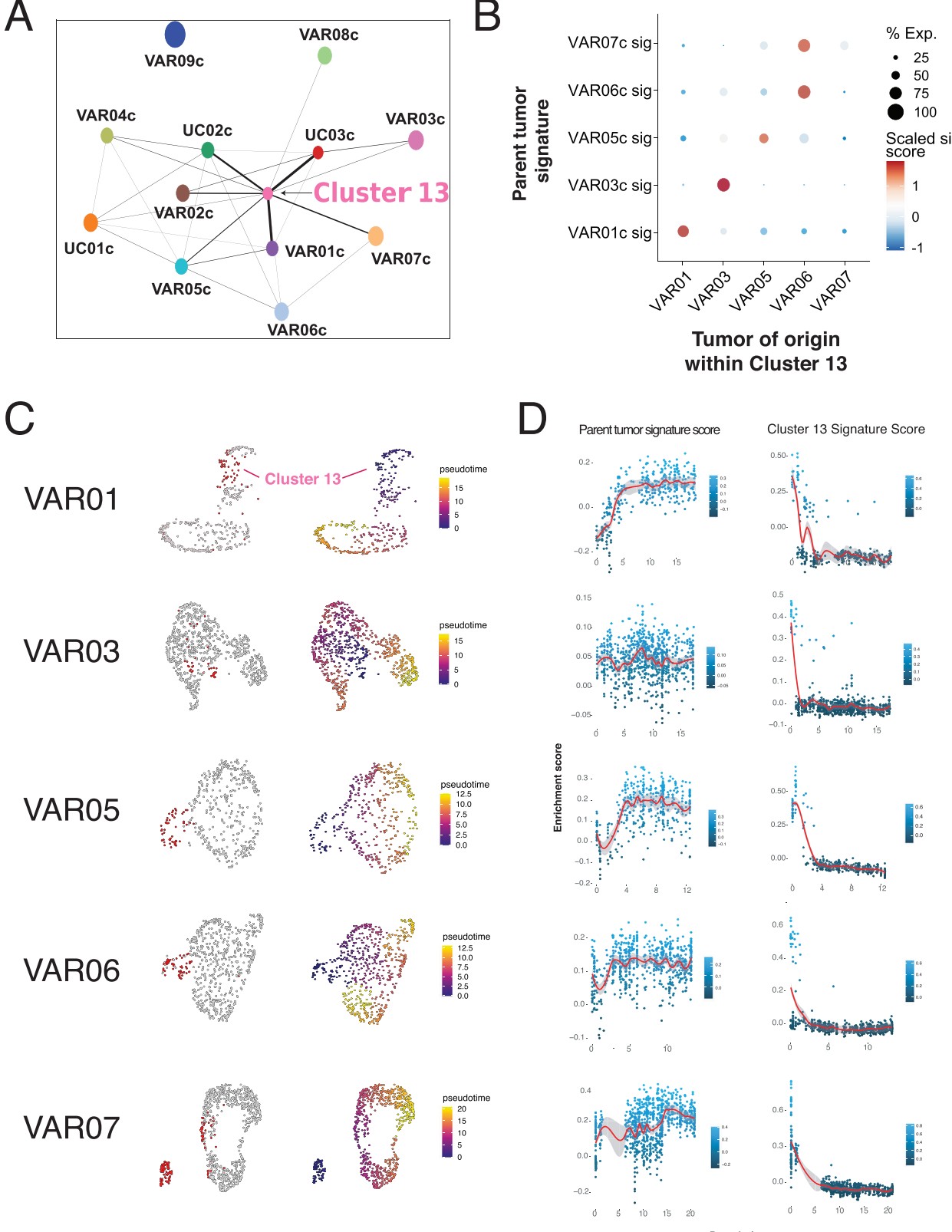

**Fig. 2 | Transcriptional relationship between Cluster 13 and parent tumor cells.**
**A** Partition-based graphical abstraction of tumor cell clusters. **B** Dot plot of tumor signature scores relative to Cluster 13 tumors of origin. **C** UMAP of individual tumors color-coded by Cluster 13 cells (red) and pseudotime using Cluster 13 cells as the starting point. **D** Expression along the pseudotime of Cluster 13 and parent tumor DEGs. Data are also presented as fitted curves with confidence intervals. Source data are provided as a Source data file.

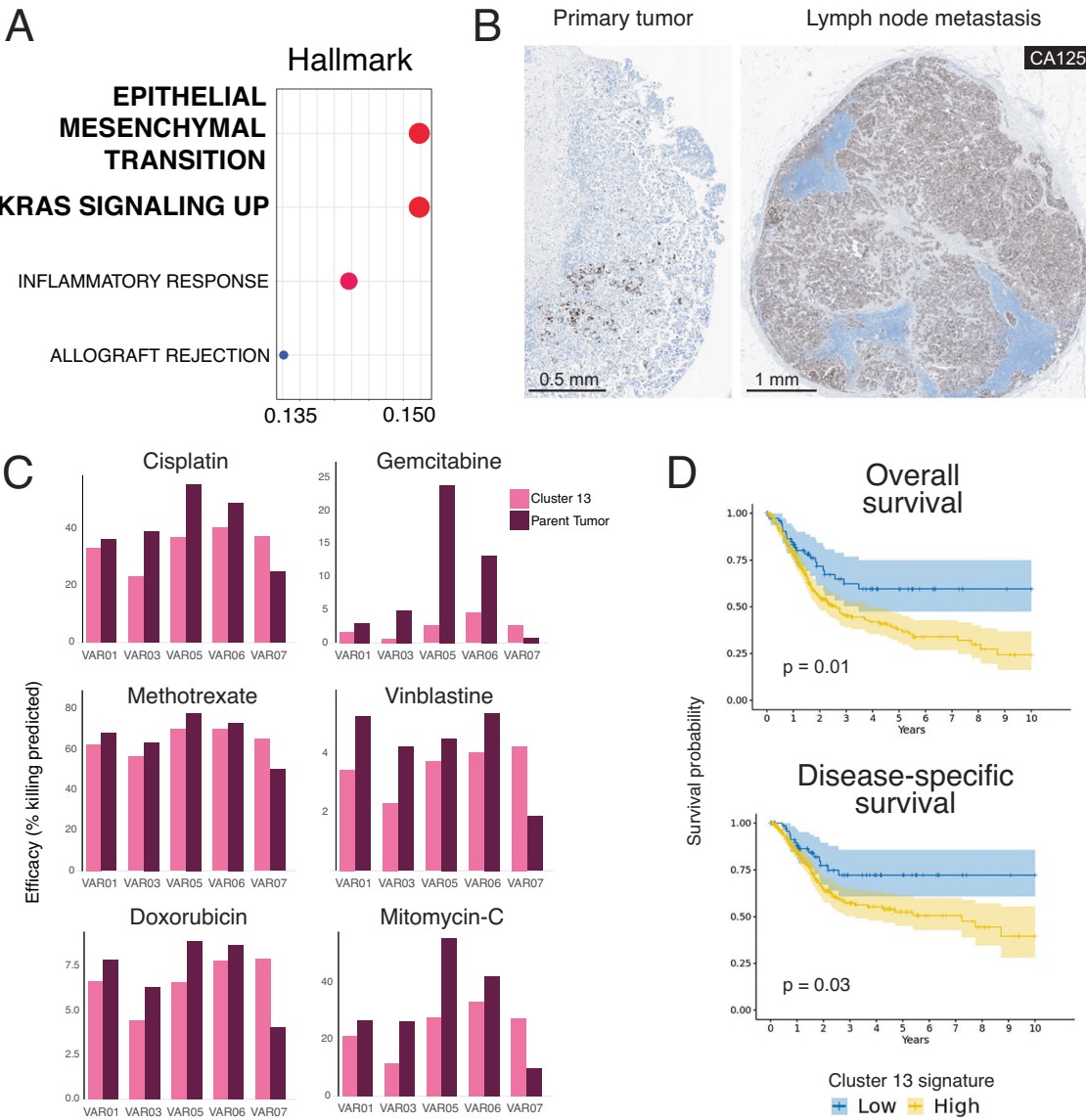

**Fig. 3 | Cluster 13 is associated with metastasis and chemotherapeutic resistance. A** Gene ontology analysis of Cluster 13 gene signature. **B** Representative CA125 immunohistochemistry in a primary HV bladder tumor and the corresponding lymph node metastasis (*N* = 4 of 5 cases examined). **C** Intratumoral comparison of Cluster 13 and parent tumor gene signature susceptibility to common bladder cancer chemotherapeutic agents in 5 HV tumors. **D** Kaplan–Meier curves of overall and disease-specific survival according to Cluster 13 signature enrichment in TCGA-BLCA (log-rank test, *p* = 0.01 for overall survival; *p* = 0.03 for disease-specific survival). Data are presented as mean values ± standard error. Source data are provided as a Source data file.

TCGA-BLCA that harbor higher Cluster 13 signature scores had significantly worse overall survival and disease-specific survival (Fig. 3D).

Taken together, these results indicate that HV tumors harbor a cancer cell state that may be more likely to metastasize and resist conventional chemotherapy. This cell state offers a potential mechanism to help explain why HV tumors are more aggressive than UC tumors.

## TM4SF1 is a surface protein broadly enriched in histologic variant tumor cells

Having identified and characterized the Cluster 13 cell state in HVs, we next asked whether our scRNA-seq results could help identify any molecular features broadly enriched in HV tumor cells compared to UCs; defining such features would facilitate the development of HV-specific targeted therapies. We categorized all tumor cells as either HV or UC according to the histology of the parent tumor and computed the DEGs (Fig. 4A). *TM4SF1*, a gene implicated in bladder

cancer as a cell cycle and apoptosis regulator, was the top DEG in the HV group[26–28]. Most HV tumor clusters, including Cluster 13, exhibited higher expression of *TM4SF1* compared to tumor clusters from pure UC tumors (Fig. 4B). VAR03c and VAR09c were the only HV tumor cell clusters with absent *TM4SF1* expression.

Consistent with previous reports, we confirmed that high *TM4SF1* expression is associated with basal tumor signatures (Supplementary Fig. 10A) and adverse clinical outcomes in TCGA-BLCA (Supplementary Fig. 10B, C)[38]. In our tumor epithelial dataset, genes with the strongest positive correlation with *TM4SF1* expression within the HV tumor cells were *EMP1*, *CLDN4*, *EZR*, and *KRT19* (Supplementary Fig. 11A, B). We checked the associations within each *TM4SF1*-expressing tumor in our scRNA-seq dataset and found these genes to be positively correlated with *TM4SF1* expression and statistically significant in each case (Supplementary Fig. 11C). *EMP1*, a gene implicated in cisplatin resistance and cancer recurrence[39–41], and *CLDN4*, a tight junction gene implicated in facilitating aggressive biology in bladder cancer[42], were also positively

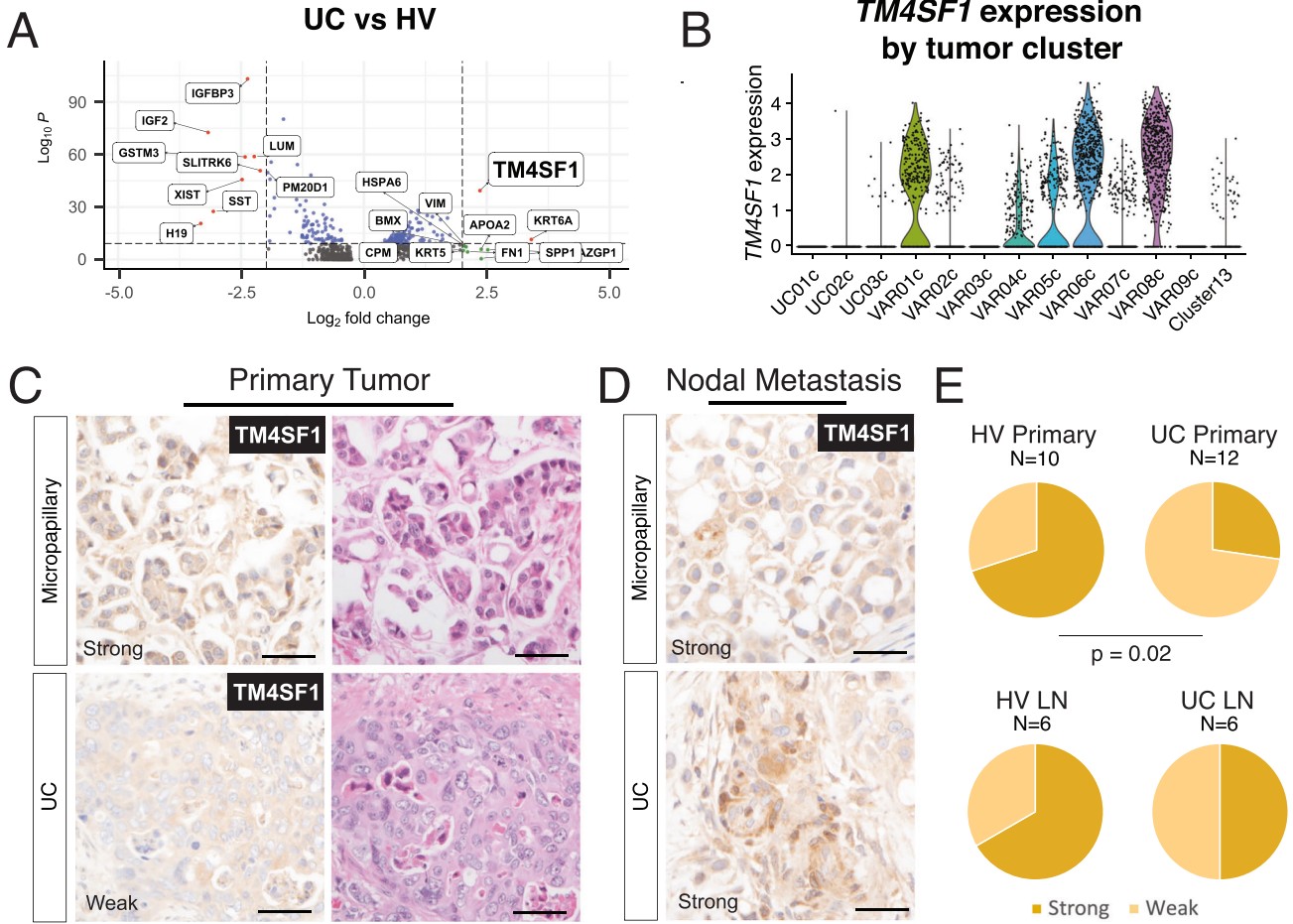

**Fig. 4 | Identification of TM4SF1 as a gene enriched in HVs. A** Volcano plot comparison of all UC and HV cells after downsampling (*N* = 150 cells per patient) (*p*-values are computed from the two-sided Wilcoxon rank-sum test). **B** Violin plots of TM4SF1 expression by tumor cluster. Source data are provided as a Source data file. **C**, **D** Immunohistochemistry of TM4SF1 in a validation cohort of HV and UC **C** primary tumors and **D** lymph node metastases. **E** Semiquantitative comparison of TM4SF1 staining in HV and UCs (Fisher's exact test, *p* = 0.02). Scale bars = 50 μm.

### Table 1 | TM4SF1 surface expression by cell line

| Cell line | Median fluorescent intensity shift | TM4SF1 expression (log2) |
|---|---|---|
| UMUC3 | 609 | 9.72 |
| 5637 | 501 | 8.21 |
| 253JBV | 451 | 10.28 |
| UMUC1 | 335 | 5.46 |
| T24 | 306 | 7.69 |
| HT1376 | 0 | 0.99 |

associated with *TM4SF1* in TCGA-BLCA (Supplementary Fig. 11D). Interestingly, we did not observe a statistically significant association between *TM4SF1* expression and *SOX2*, *DDR1*, *MMP2*, or *MMP9* expression, suggesting that the expression of *TM4SF1* in HVs may be regulated differently than what has been previously described in cell lines and non-urothelial cancers[43,44].

Using immunohistochemistry, we validated TM4SF1 protein expression in HV and UC cells, both in primary tumors and lymph node metastases (Fig. 4C, D). Consistent with our sequencing results, quantification of TM4SF1 staining using a binary "strong" and "weak" scoring system (see methods) demonstrated more frequent strong staining in HV primary tumors compared to UC primary tumors (*p* = 0.02) (Fig. 4E).

### TM4SF1-CAR T cells demonstrate in vitro and in vivo activity against bladder cancer cells

The enrichment of *TM4SF1* expression in HVs and its cell surface localization made it a compelling candidate for developing a targeted therapeutic strategy. Expression of TM4SF1 is high across a number of tumor types, and its inverse correlation with *PVRL4* (NECTIN4) expression in TCGA-BLCA and CCLE (Supplementary Figs. 11D and 12) suggests that TM4SF1-directed therapies might be complementary to enfortumab vedotin (EV) therapy, an antibody-drug conjugate that targets NECTIN4 that was recently approved for frontline treatment of patients with locally-advanced/metastatic urothelial cancers[45,46].

Given that there are no FDA-approved TM4SF1-directed therapeutic agents, we next asked whether TM4SF1 could be targeted by chimeric antigen receptor (CAR) T cell therapy. To test this, we utilized a previously published TM4SF1 single-chain variable fragment (scFv) binder and incorporated this into a 41BB-based CAR bone in two configurations (VH-VL (CAR1) and VL-VH (CAR2)). We tested both CAR T candidates against six bladder cancer cell lines with variable levels of endogenous *TM4SF1* mRNA expression and surface protein expression (Table 1). Whereas the TM4SF1-CAR T cells exhibited anti-tumor activity against bladder cancer lines expressing TM4SF1 (including UMUC3, T24, 5637, 253JBV and UMUC1), the TM4SF1-CAR T cells did not kill HT1376, which is negative for TM4SF1 (Fig. 5A). We also found CAR1 had slightly better activity in vitro. To validate the specificity of

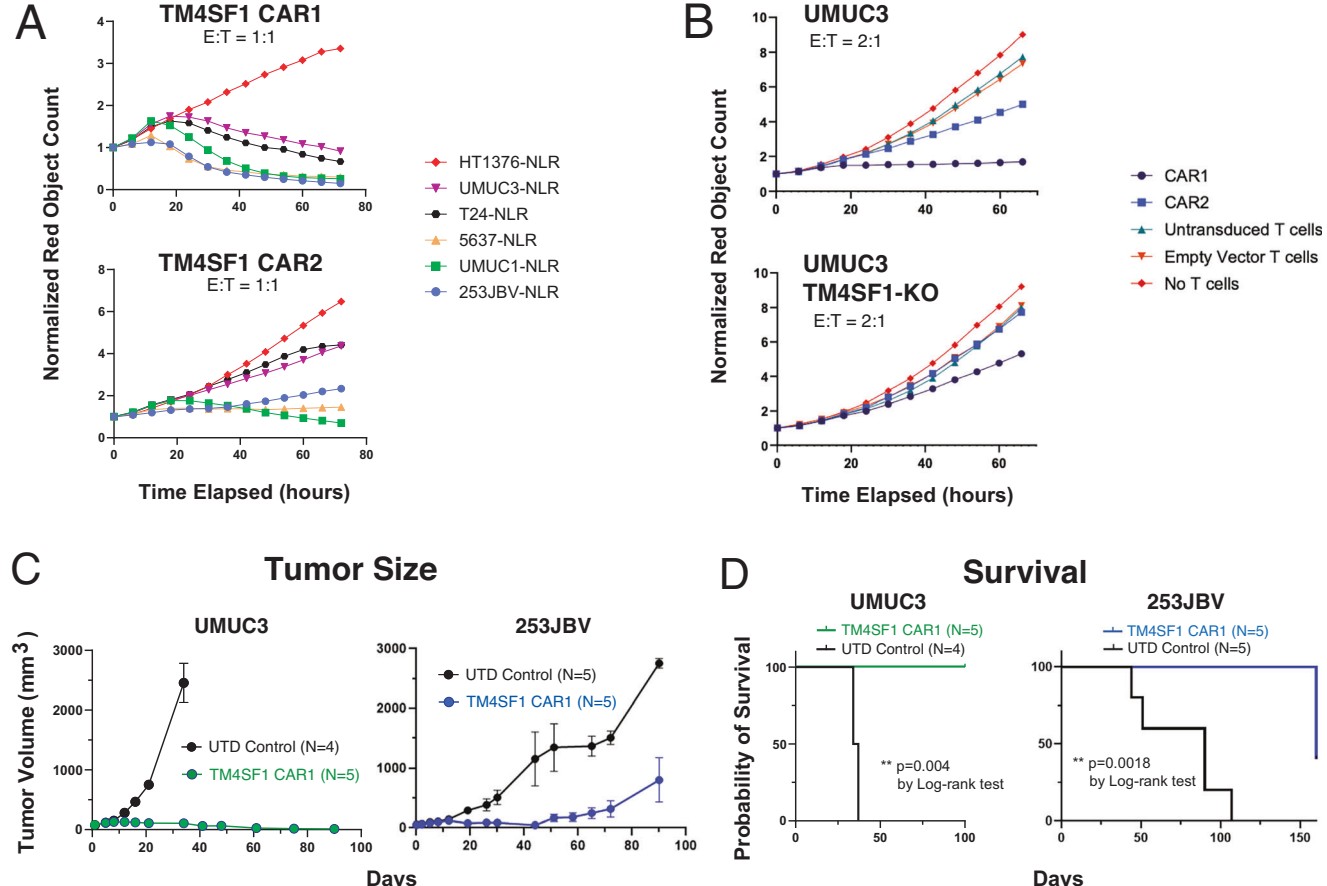

**Fig. 5 | Efficacy of TM4SF1 CAR T cells in vivo and in vitro. A** Quantification of in vitro TM4SF1-CAR1 and CAR2 activity against bladder cancer cell lines using IncuCyte co-culture assay with 1:1 effector:target cell ratio. **B** In vitro co-culture assays comparing CAR activity (2:1 effector:target cell ratio) in UMUC3 cell lines with and without CRISPR-mediated TM4SF1 knockout ($N = 1$ replicate per condition, 5 conditions per experiment). **C** Evaluation of CAR1 tumor-killing efficacy in an in vivo xenograft model using UMUC3 ($N = 9$ mice) and 253JBV ($N = 10$ mice) cell lines. Data are presented as mean values ± standard error bars. **D** Kaplan–Meier survival analysis of CAR1 treated and untreated mice with UMUC3 ($N = 9$ mice) and 253JBV ($N = 10$ mice) tumors. E:T Effector:Target, UTD untransduced, CAR chimeric antigen receptor.

our CARs, we used CRISPR/Cas9 to generate *TM4SF1* knockouts (KO) in the UMUC3 cell line, which eliminated the surface expression of TM4SF1 protein (Supplementary Fig. 13A, B) and impaired the anti-tumor activity of TM4SF1-CAR T cells (Fig. 5B).

Finally, we tested CAR1 against xenografts derived from the UMUC3 and 253JBV cell lines, which were selected for their high TM4SF1 expression and absent NECTIN4 expression. We found that CAR1 exhibited potent anti-tumor activity against these tumors in vivo (Fig. 5C). Whereas all untreated control UMUC3 mice died by day 37, all TM4SF1-CAR1-treated UMUC3 mice had a complete and durable response through day 100 (Fig. 5C, D). In the 253JBV cohort, all mice in the untransduced control group died by day 107; of the mice that received TM4SF1-CAR1 treatment, three of five (60%) survived through day 160 before reaching tumor size endpoint while two (40%) remained tumor-free through day 199 (Fig. 5D). Importantly, mice treated with TM4SF1-CAR1 cells had stable weights (Supplementary Fig. 14) and no overt pulmonary toxicity.

Taken together, these data demonstrate that TM4SF1 may be a new therapeutic target for TM4SF1-expressing bladder cancers, including tumors with variant histology and those lacking NECTIN4 expression, and can be successfully targeted using CAR T cell therapy in a mouse xenograft model. Human toxicity remains a concern, given the expression of TM4SF1 in other human tissues (Supplementary Fig. 15), so further studies are needed to characterize and mitigate these potential effects.

## Discussion

In this study, we used scRNA-seq analysis to identify shared molecular features for HV bladder cancers, a rare, understudied type of cancer. We detected a cancer cell state (Cluster 13) with clinical and mechanistic significance and found enriched expression of a targetable protein (TM4SF1) in HV bladder cancers. As HVs are poorly understood in part because they are heterogenous and uncommon, scRNA-seq enabled us to derive insights about HV cancer biology in a relatively small cohort of tumors. Our study underscores the potential of scRNA-seq technologies in precision cancer medicine[47].

The identification of a distinct "Cluster 13" cell state has important clinical implications for HV bladder cancers. This cell state is characterized by the expression of several genes that can be leveraged as biomarkers, including CA125. Indeed, CA125+ cells can be identified in most primary HV tumors and metastases. A deeper characterization of this cell state may lead to new unified strategies to treat tumors that otherwise exhibit a great degree of heterogeneity. Although these cells are predicted to be more resistant to conventional chemotherapeutics used for bladder cancer such as cisplatin, gemcitabine, doxorubicin, vinblastine, and mitomycin C, several United States Food and Drug Administration (FDA)-approved agents including omipalisib (PI3K/mTOR inhibitor) and quizartinib (FLT3 inhibitor) were predicted to be more effective against this group of cells (Cluster 13) compared to other tumor cells (Fig. S7).

The expression of *MUC16* (CA125) and other mucin genes in this cell state is intriguing. CA125, a well described gene more commonly associated with ovarian and pancreatic cancers[48–51], is a membrane-bound mucin protein that can promote cancer invasion and metastasis, and it has also been associated with therapeutic resistance in bladder cancer[52–54]. It will be important to establish in future studies whether CA125 alone contributes to HV biology and clinical behavior. Serum CA125 levels have long been used for the clinical surveillance of ovarian cancer and may have diagnostic and prognostic implications for other cancers[55,56]. Here we show that patients with HV tumors have higher serum CA125 levels compared to patients with UC tumors, supporting its potential use as a biomarker in bladder cancer and for serological detection of HV tumors.

The origin of the cancer cell state identified in Cluster 13 remains an important question. Our data suggest that Cluster 13 is a convergent cell state that could represent an epithelial-mesenchymal transition event within different tumors. However, it should be acknowledged that the temporal relationship between Cluster 13 and other cells within the parent tumor cannot be fully determined using scRNA-seq alone. How Cluster 13 relates to UC tumors is another open question. Our results do not exclude the existence of Cluster 13 in UC tumors; in fact, there may be similar convergent tumor cell behavior in UCs, albeit less frequent compared to HVs. The existence of a common cell state associated with metastasis and chemotherapy resistance among diverse tumors suggests that a common mechanism may in part underlie clinically aggressive behavior. Investigating how this cell state behaves functionally and how it arises will further inform our understanding of bladder cancer evolution and metastasis.

Finally, our discovery that most HV tumors exhibit enriched expression of *TM4SF1*, a gene that encodes a surface protein that has already been implicated in the pathogenesis of aggressive bladder cancers and other cancer cell types, has therapeutic implications[38,43,44,57]. *TM4SF1* is a promising target because its expression is not limited to HV bladder cancers and its negative association with *PVRL4/PRR4* expression suggests that targeted therapy against *TM4SF1* could complement existing targeted agents. Antibody-mediated inhibition of *TM4SF1* has been previously shown to have therapeutic potential against cancer stem cells in vitro[58]; we now demonstrate durable anti-tumor responses in mice bearing xenografts with minimal toxicity. Our preclinical testing of TM4SF1-CAR T cells thus lays a foundation for future clinical trials in bladder cancer and other tumor types expressing TM4SF1.

The primary limitation of our scRNA-seq dataset is the relatively low cell capture rate and the low sample size for UC tumors. While this is a known limitation of the Seq-Well platform and there was variable quality and viability of tumor specimens collected during surgery, we had sufficient cell numbers to investigate tumor epithelial cells. Our ability to compare differences in the tumor microenvironment and identify intercellular interactions, however, was limited. To address the low sample size of sequenced tumors, we used an existing scRNA-seq dataset as well as the TCGA-BLCA dataset to supplement our analyses. We also acknowledge that conclusions drawn from in silico assays (e.g., chemotherapy resistance) will need to be biologically validated in vitro or in vivo.

In conclusion, our study demonstrates that HVs harbor a clinically significant CA125+ cell subpopulation and that HVs are also enriched in expression of a surface antigen that is targetable using CAR T cells. These findings lay a foundation for further translational investigation into these rare, poorly understood tumors.

## Methods
### Sample collection
All bladder tumor samples used in our analyses were obtained in compliance with ethical regulations under University of California San Francisco IRB 10-04057. Informed consent was obtained from all participants or their legally authorized representatives prior to surgery. In patients undergoing transurethral resection of bladder tumor (TURBT), specimens were obtained using cold biopsy forceps. In patients undergoing radical cystectomy, specimens were obtained immediately upon removal of the bladder to minimize the effects of ischemia. Visible tumor was excised from the specimen after the bladder was opened according to standard pathology protocol. To preserve the integrity of the tissue for clinical pathology, no more than 0.5 g tumor material was taken for research. This limit was not exceeded for any subjects. All tissues were immediately placed in RPMI 1640 media on ice. Clinical and pathological data are shown in Supplementary Table 1.

### Tissue dissociation
Mechanical tissue dissociation was performed using scissors and enzymatic dissociation was performed using 1000 U/mL Type IV collagenase (Worthington, Cat: LS004188) at 37 °C for 30 min. A single-cell suspension was isolated using a 40 μm strainer, pelleted at $300 \times g$, and reconstituted in RPMI 1640 media with 10% FBS. Viability and concentration were determined using acridine orange/propidium iodide on a LUNA automated cell counter (Logos Biosystems). The suspension was then adjusted for a target loading concentration of ~50,000–100,000 live cells/mL.

### Single-cell RNA sequencing
cDNA library preparation was performed using the Seq-Well platform as previously described[59,60]. Briefly, 10,000–20,000 cells were loaded onto a Seq-Well array containing 110,000 barcoded mRNA capture beads (ChemGenes, Ct: MACOSKO-2011-10(V+)). Arrays were sealed using a polycarbonate membrane (Sterlitech, Cat: PCT00162X22100) at 37 °C for 40 min. Cells were then lysed in lysis buffer (5 M guanidine thiocyanate, 1 mM EDTA, 0.5% sarkosyl, 1% BME) for 20 min at room temperature. Hybridization of mRNA to the beads was performed in hybridization buffer (2 M NaCl, 4% PEG8000) for 40 min. The beads were then collected and washed with 2 M NaCl, 3 mM $MgCl_2$, 20 mM Tris-HCl pH 8.0, 4% PEG8000.

Reverse transcription was then performed using Maxima H Minus Reverse Transcriptase (ThermoFisher, Cat: EP0753) in Maxima RT buffer, PEG8000, template switch Oligo dNTPs (NEB, Cat: No447L), and RNase inhibitor (Life Technologies, Cat: AM2696) at room temperature for 15 min and then 52 °C overnight. Second strand synthesis was performed using Klenow Exo- (NEB, Cat: M0212L) in Maxima RT buffer, PEG8000, dNTPs, and dN-SMRT oligo for 1 h at 37 °C. Whole transcriptome amplification was performed with KAPA HiFi Hotstart Readymix PCR kit (Kapa Biosystems, Cat: KK2602) and SMART PCR Primer (Supplementary Data). The reactions were purified using SPRI beads (Beckman Coulter) at 0.6X and then 0.8X volumetric ratio.

Libraries were prepared using 800–1000 pg of DNA and the Nextera DNA Library Preparation Kit. Dual-indexing was performed using N700 and N500 oligonucleotides. Library products were purified using SPRI beads at 0.6X and then 1X volumetric ratio. A final 3 nM dilution was prepared for each library and sequenced on an Illumina NovaSeq S4 flow cell.

### Sequencing and alignment
Sequencing results were returned as paired FASTQ reads. These paired FASTQ files were then aligned against the hg19 reference genome (GRCh37.p13) using the dropseq workflow (https://cumulus.readthedocs.io/en/latest/drop_seq.html). The alignment pipeline output for each pair of FASTQ files included an aligned and corrected bam files, a digital gene expression (DGE) matrix text which was used for downstream analysis, and text-file reports of basic sample qualities such as the number of beads used in the sequencing run, total number of reads, alignment logs.

## Single-cell quality control and clustering analysis

Cells were clustered and analyzed using Seurat (v4.3.0) in R (v.4.3.1). Cells with fewer than 300 genes, 500 transcripts, or a mitochondrial gene content of 20% or greater were removed. Doublets were removed using DoubletFinder (v.2.0.3). UMI-collapsed read-count matrices for each cell were used for clustering analysis in Seurat. We followed a standard workflow by using the "LogNormalize" method that normalized the gene expression for each cell by the total expression, multiplying by a scale factor 10,000. To identify different cell types, we computed the standard deviation for each gene and returned the top 2000 most variably expressed genes among the cells before applying a linear scaling by shifting the expression of each gene in the dataset so that the mean expression across cells was 0 and the variance was 1. Principal components analysis (PCA) was run using the previously determined most variably expressed genes for linear dimensional reduction and the first 100 principal components (PCs) were stored, which accounted for 47.04% of the total variance. For graph-based clustering, the top 75 PCs and a resolution of 0.5 were selected, yielding 36 cell clusters. Differentially expressed genes (DEGs) in each cluster were identified using the FindAllMarker function within the Seurat package and a corresponding p-value was given by the Wilcoxon's rank-sum test followed by an FDR correction. In the downstream analysis, tumor cells from each patient were further clustered in a similar manner. For the individual patient clustering analysis, the number of PCs was determined by the statistical permutation test and the straw plot, and clustering resolution was selected accordingly.

## Cell-type annotation and copy number variation

To annotate each cell type from the previous clustering, we referred to canonical markers and signature gene sets developed from established studies for each cell type. We computed the signature scores of these established gene sets for each cell in our dataset using the AddModuleScore function in Seurat. Each cluster in our dataset was assigned with an annotation of its cell type by top signature scores within the cluster. To validate the identities of the tumor cell populations, we estimated copy number variants (CNV) via InferCNV (Version 1.4.0), using all the non-tumor populations as reference. During the inferCNV run, genes expressed in fewer than five cells were filtered from the dataset and the cut off was fixed at 0.1. Hidden Markov model (HMM) based CNV prediction was generated and estimated CNV events were shown in a heatmap.

## Pseudotime analysis

To further investigate the differential trajectories of tumor cells in each patient, we conducted a pseudotime analysis in Monocle3. To analyze gene expression relative to the Cluster 13 cell state, Cluster 13 cells were selected as the starting point for the pseudotime trajectory. Pseudotime trajectories were computed accordingly and visualizations were made to illustrate specific gene expression levels along the pseudotime trajectory in each patient.

## Gene ontology and gene set enrichment analysis

Within the tumor cells, we created a customized gene set signature for each variant tumor cell population of interest. Using the DEGs obtained from FindAllMarker function, we included genes with log2 fold change >2 and statistical significance (FDR $q < 0.05$) in the customized signature gene set.

To assess the in silico functional roles of Cluster 13 cells, we used the signature gene sets derived from the scRNA-seq data to run gene ontology (GO) analysis against known signature gene set collections such as Hallmark, C2CP, C2CGP, C5GO and C6 oncogene collections (https://www.gsea-msigdb.org/gsea/msigdb/). The gene ratio and statistical significance levels from the overexpression test were calculated. Normalized gene expression data and variant tumor types as metadata were used in the GSEA analysis run on the GSEA software[61,62].

To examine the association between signature gene sets or marker expression derived from our dataset and known basal/luminal signatures or canonical marker expression in the TCGA-BLCA bulk RNA sequencing dataset for validation, we performed ssGSEA (single set Gene Set Enrichment Analysis) by projecting the TCGA sample expression data onto the transcriptomic space defined by marker expression and established signature gene sets. For each target marker expression of target signature gene set, association was quantified via IC (information coefficient) and statistical significance was computed.

## CaDRReS-Sc drug response prediction

Using the tumor cells from each patient (pure UC and variant) and the subsets of Cluster 13 separated by each histological variants (Cluster13_VAR01, Cluster13_VAR03, Cluster13_VAR05, Cluster13_VAR06 and Cluster13_VAR07), we implemented the CaDRReS-Sc method[37,63] and used the CCLE cell line transcriptomic data ($N = 941$) and the updated GDSC2 database to train a model to predict the drug response for each cell cluster. A total of 297 drugs were used in the model training. "Cadres-wo-sample-bias-weight" was selected for the objective function, which is the CaDRReS-Sc model option. 500,000 steps of training were run until the mean squared error (MSE) remained unchanged in 5000 steps. Then this model was used to infer the drug response (percentage of efficacy) for each tumor cluster in our dataset.

## Survival analysis

Within the TCGA-BLCA bulk RNA-seq dataset, the Cluster 13 signature score was computed on the normalized gene expression data for each sample. Samples were then divided into high and low groups based on the 20% percentile cutoff of the Cluster 13 signature score. The overall survival (OS) distribution of both groups was compared by means of log-rank tests using the survfit function from the survival package (v3.3-1). The Kaplan–Meier (KM) survival curve was plotted using the survminer (v0.4.7) package.

## Histology and immunohistochemistry

FFPE bladder cancer tissue banked under IRB 10-04057 was sliced to 4 μm and mounted on positively charged Superfrost microscope slides. Hematoxylin and eosin (H&E) staining was performed using a standard method. CA125 (Signet, clone OC125) immunohistochemistry (IHC) was performed on an automated Ventana Benchmark Ultra IHC system using CC1 cell conditioning solution. MUC4 (CellMarque, 406M-15) and KRT24 (ThermoFisher, MA5-26581) IHC were performed at 1:50 and 1:5000 dilution, respectively, with 30-min antigen retrieval in EDTA pH 9.0 at 100 °C; TM4SF1 IHC was performed using a rabbit polyclonal antibody (Abcam, ab113504) at a 1:500 dilution after a 10-min citrate antigen retrieval at 100 °C on a Leica Bond III platform. A tissue microarray including pancreas, vascular endothelium, adipose, and lymphoid tissue was used for control. For TM4SF1 IHC, tumor staining was compared with that of endothelial cells on the same slide; tumor cells that stained equally or darker than endothelial cells were scored as "strong" while those that stained lighter were scored as "weak." All stains were reviewed by a pathologist.

## CA125 serology

Serum CA125 levels were prospectively measured in patients undergoing TURBT or cystectomy for bladder tumors using the Abbott Architect Chemiluminescent Microparticle Immunoassay (CMIA) and reviewed under IRB 10-04057. Blood samples were drawn in the preoperative area prior to surgery. Pathologic diagnoses were reviewed. Tumors with >5% HV components were categorized as "HVs" while tumors with no mention of HV were categorized as "UC." Tumors with equivocal or negligible HV components were excluded from the analysis; patients with "no tumor" on final pathology were also excluded.

## CAR constructs

The heavy (VH) and light (VL) chains of the TM4SF1 scFv binder was obtained from antibody AGX-A01 (patent US011208495B2). The VH and VL sequences were cloned in two configurations using the Gibson Assembly protocol (Twist) into a CAR backbone containing IgG4 spacers, CD8 hinge and transmembrane domain, 4-1BB costimulatory domain, CD3ζ chain, and EGFP. Plasmids were prepped using the NucleoBond Xtra Midi Plus kit (Takara Bio).

## CAR lentivirus production

For TM4SF1-CAR lentivirus production, HEK293T-Lenti-X cells (Takara Bio) were thawed, cultured, and expanded in DMEM media supplemented with 10% FBS. HEK293T-Lenti-X cells were transfected with the TM4SF1-CAR lentiviral plasmid and the packaging plasmids psPAX2 and pVSVG using the TransIT-LT1 transfection reagent (Mirus Bio). Cell supernatant was collected at 48 h and 72 h. The virus was filtered and concentrated using the Lenti-X Concentrator (Takara Bio) according to manufacturer's instructions and resuspended in serum-free media.

## TM4SF1-CAR T generation

Human T cells were isolated from a leukopak (Stemcell Technologies) using the Easy Sep Human T cell enrichment kit (Stemcell Technologies). T cells were then plated on retronectin coated plates (Takara, T100A), stimulated with Human CD3/CD28 T Cell Activator (Stemcell Technologies, 10971), and concentrated lentivirus was added. Cells with virus were spun at 1000 rpm for 45 min. After 72 h of incubation, virus was removed, and cells were allowed to recover for 2–3 days. Transduction efficiency was evaluated via flow cytometry by GFP expression. If less than 30% of the T cells were GFP positive, the cells were MACs sorted using a biotinylated c-myc antibody (Miltenyi Biotec, 130-124-877) and isolated using the MiniMACS separator and columns (Miltenyi) according to manufacturer's protocol. The CAR-T cells were grown in either ImmunoCult-XF T Cell Expansion Medium (Stemcell Technologies, 01981) or TexMACS™ Medium (Miltenyi Biotech, 130-097-196). Human recombinant IL-15 (Stemcell Technologies, 78031) and IL-7 (Stemcell Technologies, 78053), 10 ng/mL final concentration each was freshly added to the cells every 2–3 days, with cells grown at a concentration of $1 \times 10^6$ per mL and used between day 14–20 for downstream assays.

## Cell culture

5637 cells were obtained from the UCSF Cell Culture Facility. UMUC-3 cells were a gift from Bradley Stohr (UCSF). T24, UMUC-1 and 253JBV cells were gifts from Peter Black (University of British Columbia) and David McConkey (Pathology Core, Bladder Cancer SPORE, MD Anderson Cancer Center). Cells were grown in standard MEM media (Corning) supplemented with 10% FBS (Seradigm) and penicillin/streptomycin. All experiments were conducted within 20 passages from the parental stock. Cells were validated by STR profiling and routinely tested for mycoplasma (Lonza).

## TM4SF1 knockout cells

UMUC-3 TM4SF1-KO cells were generated by transient transfection (Lipofectamine 3000) of UMUC-3 cells with PX458 (Addgene, #48138). Each plasmid contained one of three different sgRNA targeting sequences: (1) AGTGCACTCGGACCATGTGG; (2) GGTGTAGTTCCACTGGCCGA; (3) ATTAGCCGCGATGCACAGGA. 48–72 h after transfection, GFP-positive cells were sorted by FACS (BD Fusion) and expanded. Cells were then stained with a TM4SF1 antibody (Miltenyi, clone REA851, 1:100), sorted a second time by FACS (BD Fusion), and negative cells were collected and expanded.

## TM4SF1 flow cytometry

Flow cytometric quantification of TM4SF1 expression across human bladder cancer cell lines was performed by incubating with anti-TM4SF1-PE antibody (Miltenyi, clone REA851, 1:100) for 30–60 min on ice. Cells were analyzed using an Attune NxT Flow and the median fluorescence intensity (MFI) was calculated and data were analyzed using FlowJo software.

## IncuCyte co-culture assays

Bladder cancer cells labeled with NucLightRed (Sartorius) were co-cultured with human non-transduced (NTD) T cells or TM4SF1-CAR T cells at variable effector-to-target (E:T) ratios. On day 0, 2000–5000 target cells were plated and allowed to adhere overnight. On day 1, effector T cells were added and tumor cell killing was monitored on an IncuCyte S3 (Sartorius). Images were obtained every 3–6 h over 72–96 h. Target cells were quantified based on the red object count or red area confluence normalized to the starting day 1 values, and data were plotted on Prism (GraphPad, v10).

## Animal studies

All animal studies were performed under an approved UCSF Institutional Animal and Use Committee (IACUC) protocol 194778. NSG (NOD/SCID/gamma) mice were housed in the UCSF barrier facility under pathogen-free conditions and were obtained through an in-house breeding core. For subcutaneous xenografts, $1 \times 10^6$ cells were injected into the left flank of 8–10-week-old male NSG mice. The injected cells were resuspended in 1:1 serum-free media and Matrigel (BD Biosciences). Mice were enrolled into treatment groups once tumor volumes reached between 50 and 100 mm$^3$, typically 10–14 days after tumor cell inoculation. An intravenous injection of $3–5 \times 10^6$ untransduced (UTD) control or TM4SF1 CAR T cells was then delivered through the tail vein. Tumors were measured with digital calipers and mice were weighed twice weekly by personnel from the UCSF Preclinical Therapeutics Core in a blinded fashion. Tumor volumes were recorded using Studylog Animal Study Workflow software and plotted using Prism (GraphPad, v10). Mice were euthanized when tumors reached 20 mm in any direction. For survival analysis, a log-rank test was used to compare the overall survival of mice in each cohort.

## Statistics and reproducibility

This study was designed to compare cases (HV tumors) with controls (UC tumors) using single-cell analysis. No statistical method was used to pre-determine overall sample size. The experiments were not randomized, and the investigators were not blinded to allocation during experiments and outcome assessment. Sample size calculation was not used to pre-determine the number of tumors sequenced, but we applied quality control metrics (viability, cell count) to determine which tumors would be included in our dataset. Of the 15 tumors that were sequenced, two HV tumors and one UC tumor was excluded from our primary analyses due to low capture rate of tumor epithelial cells ($N < 150$). Blinding was not relevant to the descriptive computational analyses and therefore not performed. Animal CAR T experiments were blinded as described above. Sample sizes for CAR T experiment cohorts were determined based on expected effect size and threshold for achieving biological replicants. Data are shown as mean ± standard deviation (if normally distributed), mean and interquartile range (IQR) (25th–75th percentile; if not normally distributed), or point values. Group comparisons were performed using Student's t-test, Wilcoxon rank-sum test, log-rank test, or Fisher's exact test. Statistical significance for all tests was set at $p < 0.05$.

## Reporting summary

Further information on research design is available in the Nature Portfolio Reporting Summary linked to this article.

# Data availability

Raw single-cell RNA sequencing FASTQ files and gene expression matrices files generated in this study have been deposited in the Gene

Expression Omnibus (GEO) and are publicly available with the accession number GSE293189. The remaining data are available within the Article or Supplementary Information. Source data are provided with this paper.

## Code availability

All software algorithms used for analysis are available for download from public repositories. All R code used to generate figures in the manuscript are available at https://github.com/angelussong/Histological_Variant_Bladder_Cancer_Analysis.

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

## Acknowledgements

This project was supported by the Chan Zuckerberg Biohub San Francisco (F.H.), the UCSF Department of Medicine (F.H.), TL1DK139565 (H.Y.), U2CDK133488 (H.Y.), the Urology Care Foundation (H.Y.), and the California Urology Foundation (H.Y.). We thank Alex Shalek for support of the Seq-Well platform. We thank all members of the Huang lab for their input and support.

## Author contributions

H.Y., H.S., S.P., and F.H. conceptualized the research framework and hypotheses. S.P. and M.M. provided clinical samples for analyses. H.Y., H.S., and P.A. generated the scRNA-seq dataset. H.Y., H.S., and J.G. performed computational analyses. H.Y., T.G., K.L., B.S., and C.D. performed histological and immunohistochemistry analyses. C.K. and A.W. generated CAR constructs. E.Y., K.C., K.H., J.P., C.K., V.K., and J.C. performed and analyzed in vitro and in vivo CAR T experiments. H.Y. and H.S. wrote the initial draft of the manuscript, which was then edited by all authors. S.P. and F.H. supervised the overall project. F.H. provided research funding. H.S. and F.H. will be responsible for curating and maintaining the integrity of all data.

## Competing interests

J.C. and F.H. are inventors on a patent application related to the CAR T cell technology described in this manuscript ("TM4SF1 CAR CELLS AND METHODS OF USE THEREOF", U.S. Patent Application No. 63/649,157, U.S. Patent Application No. 63/649,821). The remaining authors declare no competing interests.
