## [Transparent Peer Review file · Nature Communications]

Bladder cancer variants share aggressive features including a CA125+ cell state and targetable TM4SF1 expression

Corresponding Author: Professor Franklin Huang

Version 1:

Reviewer comments:

Reviewer #1

(Remarks to the Author)

This study compares a small number of histological variant subtypes of bladder cancer with transitional cell cancers. HV are a very heterogenous group of cancers with a broad prognostic spectrum and it is difficult to know how to translate these findings into a clinical setting. Further validation of these findings are required in clinical datasets to understand the significance of these findings. Are there any such datasets available as this greatly strengthen the study?

Introduction

No comments. Well written and clear introduction to the topic

Methodology

No comments. Clear methodology that allows a good understanding of how the distinct approaches and analyses were performed.

Results

“To validate the existence of Cluster 13 cells histologically, we performed immunostaining of CA125 (encoded by MUC16) in HV (N=14) and UC (high-grade invasive and carcinoma in situ) tumors (N=20). We found a subpopulation of CA125+ cells in a variety of HV tumors with different subtypes (13/14) (Fig. 1E) but rarely in tumors with UC (1/11) or carcinoma in situ (CIS) histology (1/9).”

IHC validation could be improved: Only CA125 staining was attempted, which does not represent the totality of the cluster. While it is a good proof-of-concept experiment, additional markers (e.g. KER24) should have been included for more definitive IHC validation. In that sense, if CA125 was chosen as a proxy, it should be stated why it was selected over its counterparts.

“While scRNAseq alone cannot prove the temporal relationship between these cells, our results support the idea that cancer cells found in Cluster 13 are a convergent cell state in HV tumors.”

If Cluster 13 is convergent for HV tumours only, why it associated with UCs tumours? I understand it can be a convergent cell state which is more prevalent/maintained in HV tumours, but that fact should be clearer in the writing to avoid contradictions in the writing.

“The Cluster 13 signature was anticorrelated with the parent tumor signature in four of five tumors (Fig. 3D), and the marked contrast of the Cluster 13 signature along the pseudotime in all five tumors suggests that Cluster 13 arises as a derivative of the parent tumor rather than vice versa.”

Why only five tumours were used for the analysis? Were UC or HV tumours? The results makes sense but I feel context is missing regarding the tumours used for this experiment.

“Using CA125 again as a putative marker for Cluster 13 cells, we examined CA125 staining in five HV tumors with lymph node metastases and observed a striking homogeneous enrichment of CA125+ cells in the lymph nodes compared to the primary tumor in 4 of 5 cases”

Again, a limitation of the study is the fact that CA125 was used as a putative marker for a whole cell cluster. Either more markers are needed, or a validation showing that CA125 as the putative marker is robust to represent the discoverer cell cluster. For example, when performing the prognosis analysis in the TCGA-BLCA cohort, is CA125 expression independently a prognostic marker? If yes, does it correlate well with the score derived from when using the whole Cluster

13 as a prognostic marker? If clinically both are comparable, that would provide reassurance on its use as a putative marker.

Discussion & conclusion

“The identification of a distinct “Cluster 13” cell state, which was found in more than half of the sequenced HV tumors and can be detected using MUC16 (CA125) as a marker, has potentially important clinical implications for HV bladder cancers.” Is not clear to me enough evidence is provided to claim CA125 as a putative marker for Cluster 13. While it is part of the Cluster 13 signature, there is no analysis showing CA125 has the same stratification capacity as a Cluster 13 signature, and that should be added into the study to make this claim.

“The origin of the cancer cell state identified in Cluster 13 remains an important question. While our data suggest that the Cluster 13 cell state is a shared state that is found in different HV tumors, the temporal relationship between Cluster 13 cells and other cancer cells within each tumor cannot be determined using scRNA-seq alone.”

An important point to address is the relationship predicted between Cluster 13 and UCs tumours in Figure 2A. Although it is clear that Cluster 13 is shared and more prevalent among HV tumours, and there is evidence suggesting it can be a convergent precursor state, there is also the possibility it can be associated with UCs.

(Remarks on code availability)

Reviewer #2

(Remarks to the Author)

Yang et. al. investigated transcriptional programs of the rare histologic variant (HV) subtype of bladder cancer. They identified a distinct cell state enriched in but not unique to HV (compared to UC), which is associated with a pro-metastatic and drug-resistant expression program. Overall, the study is valuable and identifies a targetable feature of an understudied tumor subtype with poor prognosis. I also appreciated the author’s discussion regarding how scRNA-seq is limited in drawing conclusions on the origin, temporal relationships and function of this cell state. This computational analysis is valuable and lays the foundation for future mechanistic studies. I was asked to review the scRNA-seq aspects of the study. My comments are as follows:

Major:

1. The cluster 13 cell state was enriched in but not unique to HV as compared to UC. My impression is that the validation TCGA and scRNA-seq datasets represent only UC. Could the authors please discuss in a bit more detail what the resistance and survival findings mean in this context? Are they translatable to HV? Are there any public HV datasets that could be used for validation? To be clear, I am not suggesting generating such a dataset, which would be beyond the scope of this manuscript given the rarity of these tumors.
2. No across sample harmonization was used. Without this, cluster 13 is the only shared feature that is picked up on analysis. Would this change with harmonization which aims to remove sample specific batch effects? Are other stromal and immune cells shared across samples despite no harmonization? This would potentially justify not harmonizing the tumor cells.
3. The authors used InferCNV to confirm tumor cell contents. However, this result could be expanded further upon. Are there any copy number profile differences/similarities between UC vs HV/cluster 13? Could these explain biological differences between the two? Is there a possibility of copy number alteration induced gene dosage effect for genes of interest?
4. Please add details on the methods for the CaDRReS results. Was this performed on average pseudobulk expression or using the new single-cell version of the package? The citation refers to the older package. Was the pre-existing CCLE model used? If new model training was conducted, please list details of the training parameters used.

Minor:

1. Pseudotime trajectory callouts should refer to Fig. 2.
2. From Fig. 1B, it is hard to tell if all HVs were represented in cluster 13. Could the authors please revise the color scheme/present this as a table? Based on subsequent results, it seems that certain samples are not represented in cluster 13. Does this correlate with their primary/secondary histology or any other features?

(Remarks on code availability)

Reviewer #3

(Remarks to the Author)

This manuscript entitled “Histologic variants in bladder cancer share aggressive molecular features including a CA125+ cell state and targetable TM4SF1 expression” by Yang et al. studies the tumor biology of histologic variant (HV) subtypes of bladder cancer by performing single-cell RNA sequencing on 9 HV tumors and 3 common bladder cancers. Based on the single-cell RNA-seq data, the authors discovered a CA125+ cell cluster, which is specific to HVs, is responsible for the aggressive biology of HVs. Lastly, the authors demonstrated the utility of CAR-T cells engineered against TM4SF1 in treating HVs. After carefully evaluating the significance of this work, the conclusion drawn from the results, and the methodology, I suggest that this manuscript is not suitable for publication in Nature Communications. Please see my detailed comments below.

Major points:

1. The authors claimed that cluster13 is mainly contributed by HVs. This conclusion is not very convincing because: 1) the 3 bladder cancer samples contained much fewer cells than HVs (Figure S2C); 2) only 3 bladder cancer samples were used and the UC03 also contained cluster13 cells.
2. Line 547, the authors concluded that HVs harbored a special cell cluster expressing TM4SF1, which can be targeted by CAR-T cells. I find this conclusion also unconvincing. Among the 9 HVs subjected to scRNA-seq, cluster 13 cells were present in VAR01, VAR03, VAR05, VAR06, and VAR07 (Figure 1), whereas TM4SF1 was highly expressed in VAR01, VAR04, CAR05, CAR06, and VAR08 (Figure5B). It is unclear whether cluster 13 cells express TM4SF1 or if these are two distinct cell groups.
3. The authors claimed cluster 13 cells were more resistant to chemotherapy drugs based on transcriptomic analysis using an online system. It would be more convincing if the authors could isolate these cells based on the cell surface markers, (e.g., CA125) from clinical samples and test their drug responses.
4. While I understand that the authors put a lot of effort into analyzing the scRNA-seq data of VAR09 and VAR08, Figure 4 and the corresponding text (lines 372-424) were abrupt in this paper. VAR08 and VAR09 did not contain cluster 13 cells, which are the main focus of this manuscript. Additionally, drawing conclusions from a single sample of a particular histologic subtype may not be representative, especially given the disease's heterogeneity. I suggest the authors delete this part from the manuscript.
5. For CAR-T studies, T cells transfected with empty vector-lentivirus are suggested to be used as control cells instead of untransfected T cells. In Figure 6B, CAR2 cannot kill UMUC3 cells, which contradicts the results shown in Figure S14B.
6. It seems that TM4SF1-CAR-T cells only slightly suppressed the growth of UMUC3 cells in vitro (Figure 6B, Figure S14C), but completely killed the tumors in vivo, which is very unexpected.

Minor points:

1. The sample number of HVs in the analysis of serum CA125 levels is too small.
2. A picture of mice bearing tumors or resected tumors should be provided in Figure 6, and the number of mice used in each group should be indicated.
3. Scale bars were missing in Figure 5D and Figure S1, and were not annotated in the figure legends for Figure 5C.
4. The authors removed the correlation plots of TM4SF1 and EMP1, EZR, CLDN4, and KRT19 to Figure S12, but the figure legend remained in Figure 5.
5. Figure S14, TM4SF1 knockout efficiency in UMUC3 cells should be indicated.
6. Line 225, why can a biotinylated c-myc antibody be used to sort transfected cells?
7. Fonts should be unified in figures, and font size should be increased in Figure S11B.

(Remarks on code availability)

Single-cell RNA-seq files and all code used in the present study are not available in this manuscript.

Version 2:

Reviewer comments:

Reviewer #1

(Remarks to the Author)

(Remarks on code availability)

Reviewer #2

(Remarks to the Author)

My concerns have been adequately addressed by the authors.

(Remarks on code availability)

Reviewer #4

(Remarks to the Author)

This is a review for a resubmission, and I did not serve as a reviewer for the initial version. The original reviewers have provided extensive comments on the manuscript and I was specifically asked to comment on whether the authors sufficiently addressed the comments made by original reviewer #3. I am mostly restricting my comments to address that question with a focus on the main points raised by the original reviewer.

My overall assessment is that the authors have made a good effort to address the points raised by original reviewer number 3:

In my opinion major points 1-3 were sufficiently addressed by the authors. For instance, the authors included an additional data set of >6000 UC cells to strengthen the point that Cluster 13 cells are mainly specific for the HV subset.

I agree with the assessment of reviewer 3 regarding the data presented in Figure 4 (major point 4). To me this part also seems to be disconnected from the rest of the story. I would suggest to rather put this data to the supplement and to also deemphasize this part in the text.

Regarding the CAR T cell related part:

The points raised by original reviewer number 3 were sufficiently addressed in my opinion.

I have a few additional points:

1. In Figure 6E right panel the authors write in the text that some mice survived until day 199 and others had to be taken out at day 160. However, the survival curve ends at day 150. To reflect the entire picture the survival curve until day 160 should be displayed.
2. How did the 253JBV tumors escape the CAR T therapy? Did they lose target expression?
3. Do the cell lines used for the CAR T studies represent HV tumors to justify the following conclusion: "Taken together, these data demonstrate that TM4SF1 may be a new therapeutic target for HV bladder cancers, including tumors lacking NECTIN4 expression, and can be successfully targeted using CAR T cell therapy".
4. What is the expression pattern of TM4SF1 on healthy tissue? The authors did not see any signs of toxicity with these CAR T cells in mice (cross reactivity of the original antibody with murine tissue?) but it would be good to at least comment on potential on target off tumor toxicities of TM4SF1 targeting CAR T cells.

(Remarks on code availability)

RESPONSE TO REVIEWERS' COMMENTS

Reviewer #1 (Remarks to the Author):

This study compares a small number of histological variant subtypes of bladder cancer with transitional cell cancers. HV are a very heterogenous group of cancers with a broad prognostic spectrum and it is difficult to know how to translate these findings into a clinical setting. Further validation of these findings are required in clinical datasets to understand the significance of these findings. Are there any such datasets available as this greatly strengthen the study?

We thank the reviewer for this suggestion. Unfortunately, there are no curated HV datasets available at this time. This is certainly an important unmet need in the field and we hope our study will spur additional datasets on HVs in the field.

Introduction

No comments. Well written and clear introduction to the topic

Methodology

No comments. Clear methodology that allows a good understanding of how the distinct approaches and analyses were performed.

Results

“To validate the existence of Cluster 13 cells histologically, we performed immunostaining of CA125 (encoded by MUC16) in HV (N=14) and UC (high-grade invasive and carcinoma in situ) tumors (N=20). We found a subpopulation of CA125+ cells in a variety of HV tumors with different subtypes (13/14) (Fig. 1E) but rarely in tumors with UC (1/11) or carcinoma in situ (CIS) histology (1/9).”

IHC validation could be improved: Only CA125 staining was attempted, which does not represent the totality of the cluster. While it is a good proof-of-concept experiment, additional markers (e.g. KER24) should have been included for more definitive IHC validation. In that sense, if CA125 was chosen as a proxy, it should be stated why it was selected over its counterparts.

We thank the reviewer for this suggestion and agree that IHC validation would be stronger with additional markers. Therefore, we tested MUC4 and KRT24 IHC in several HV tumors. MUC4 and CA125 showed clear overlap in both cases (Fig. 1E). KRT24 staining was detected in all tumor cells, but the areas of CA125 and MUC4 positivity exhibited a stronger signal. We also attempted to perform WISP2 IHC, but the antibody could not be validated with our controls, so this marker was ultimately not used. Figure 1 has been updated to include representative images of MUC4, CA125, and KRT24 staining. The text was modified to reflect these additional studies and also to explain our rationale for using CA125 as a surrogate marker given its history of use as a tumor marker in other cancers and the robustness of the clinically used antibody.

(Lines 290-296): **“We observed close colocalization of MUC4 and CA125 in a subpopulation of tumor cells and a stronger KRT24 signal corresponding to these cells in two HVs (Fig. 1E). We then used CA125 as a surrogate marker for Cluster 13 in a larger bladder cancer cohort (14 HV tumors, 20 UC tumors) – CA125 was chosen due to its extensive history as a tumor marker in other cancers and the availability of clinically validated antibodies. We found a subpopulation of CA125+ cells in a variety of HV tumors with different subtypes (13/14) (Fig. 1E) but rarely in tumors with high-grade UC (1/11) or carcinoma in situ (CIS) histology (1/9).”**

“While scRNAseq alone cannot prove the temporal relationship between these cells, our results support the idea that cancer cells found in Cluster 13 are a convergent cell state in HV tumors.”

If Cluster 13 is convergent for HV tumours only, why it associated with UCs tumours? I understand it can be a convergent cell state which is more prevalent/maintained in HV tumours, but that fact should be clearer in the writing to avoid contradictions in the writing.

We agree with the reviewer that the original text is unclear. We have modified the text to reflect this.

(Lines 329-331): “While scRNA-seq alone cannot prove the temporal relationship between these cells, our results support the idea that cancer cells found in Cluster 13 are a convergent cell state that is more prevalent in HV tumors.”

**“The Cluster 13 signature was anticorrelated with the parent tumor signature in four of five tumors (Fig. 3D), and the marked contrast of the Cluster 13 signature along the pseudotime in all five tumors suggests that Cluster 13 arises as a derivative of the parent tumor rather than vice versa.”
Why only five tumours were used for the analysis? Were UC or HV tumours? The results makes sense but I feel context is missing regarding the tumours used for this experiment.**

We acknowledge that our rationale for selecting these 5 tumors is unclear in the original text. These were selected because these tumors had the highest Cluster 13 cell content to make our pseudotime analysis meaningful. We have modified the text to reflect our thinking.

(Lines 317-320): “For the five tumors that had the highest Cluster 13 content (VAR01, VAR03, VAR05, VAR06, and VAR07), all of which were HVs, we found that Cluster 13 cells bore the signature of the parent tumor with a high degree of specificity, supporting the likelihood that all cells within these tumors are clonally related (Fig. 2B).”

**“Using CA125 again as a putative marker for Cluster 13 cells, we examined CA125 staining in five HV tumors with lymph node metastases and observed a striking homogeneous enrichment of CA125+ cells in the lymph nodes compared to the primary tumor in 4 of 5 cases”
Again, a limitation of the study is the fact that CA125 was used as a putative marker for a whole cell cluster. Either more markers are needed, or a validation showing that CA125 as the putative marker is robust to represent the discoverer cell cluster. For example, when performing the prognosis analysis in the TCGA-BLCA cohort, is CA125 expression independently a prognostic marker? If yes, does it correlate well with the score derived from when using the whole Cluster 13 as a prognostic marker? If clinically both are comparable, that would provide reassurance on its use as a putative marker.**

We thank the reviewer for raising this important point. The additional IHC we performed with MUC4 and KRT24 described above should strengthen our rationale for using CA125 as a marker for this cell cluster. We also confirmed that MUC16 is an independent prognostic marker for overall and disease-specific survival when normalized to KRT7 expression:

Overall survival (via <https://tau.cmmmt.ubc.ca/cSurvival/>) (p = 0.03):

Disease specific survival ($p = 0.02$):

MUC16 expression also positively correlates with the Cluster 13 signature in TCGA-BLCA, which contains both UC and HV tumors:

While these results support our rationale, we acknowledge that more validation needs to be done to support the use of CA125 as a sole marker for this cell state.

Discussion & conclusion

“The identification of a distinct “Cluster 13” cell state, which was found in more than half of the sequenced HV tumors and can be detected using MUC16 (CA125) as a marker, has potentially

important clinical implications for HV bladder cancers.”

Is not clear to me enough evidence is provided to claim CA125 as a putative marker for Cluster 13. While it is part of the Cluster 13 signature, there is no analysis showing CA125 has the same stratification capacity as a Cluster 13 signature, and that should be added into the study to make this claim.

We thank the reviewer for this important critique. While we have provided additional evidence that CA125 spatially overlaps with other Cluster 13 markers, we acknowledge that more work needs to be done before we can claim that CA125 on its own can be used as a definitive marker for Cluster 13. We have thus modified the text to soften our claim:

(Lines 482-486): “The identification of a distinct “Cluster 13” cell state has potentially important clinical implications for HV bladder cancers. Cluster 13 cells are characterized by the expression of several genes that can be leveraged as biomarkers, including CA125. Indeed, CA125+ cells can be identified in most primary HV tumors and metastases. A deeper characterization of this cell state may lead to new unified strategies to treat tumors that otherwise exhibit a great degree of heterogeneity.”

“The origin of the cancer cell state identified in Cluster 13 remains an important question. While our data suggest that the Cluster 13 cell state is a shared state that is found in different HV tumors, the temporal relationship between Cluster 13 cells and other cancer cells within each tumor cannot be determined using scRNA-seq alone.”

An important point to address is the relationship predicted between Cluster 13 and UCs tumours in Figure 2A. Although is clear that Cluster 13 is shared and more prevalent among HV tumours, and there is evidence suggesting it can be a convergent precursor state, there is also the possibility if can be associated with UCs.

We thank the reviewer for this suggestion. We have included discussion of this possibility in our modified text:

(Lines 503-511): “The origin of the cancer cell state identified in Cluster 13 remains an important question. Our data suggest that Cluster 13 is a convergent cell state that could represent an epithelial-mesenchymal transition event within different tumors. However, the temporal relationship between Cluster 13 and other cells within the parent tumor cannot be determined using scRNA-seq alone. How Cluster 13 relates to UC tumors is another open question. Our results do not exclude the existence of Cluster 13 in UC tumors; in fact, there may be similar convergent cell behavior in UCs, albeit much less frequently compared to HVs. Regardless, the existence of a common cell state associated with metastasis and chemotherapy resistance among diverse tumors suggests that a common mechanism may underlie clinically aggressive behavior. Investigating how this cell state behaves functionally and how it arises may help inform our understanding of bladder cancer tumor evolution and metastasis.”

Reviewer #2 (Remarks to the Author):

Yang et. al. investigated transcriptional programs of the rare histologic variant (HV) subtype of bladder cancer. They identified a distinct cell state enriched in but not unique to HV (compared to UC), which is associated with a pro-metastatic and drug-resistant expression program. Overall, the study is valuable and identifies a targetable feature of an understudied tumor subtype with poor prognosis. I also appreciated the author’s discussion regarding how scRNA-seq is limited in drawing conclusions on the origin, temporal relationships and function of this cell state. This computational analysis is valuable and lays the foundation for future mechanistic studies. I was asked to review the scRNA-seq aspects of the study. My comments are as follows:

Major:

1. The cluster 13 cell state was enriched in but not unique to HV as compared to UC. My impression is

that the validation TCGA and scRNA-seq datasets represent only UC. Could the authors please discuss in a bit more detail what the resistance and survival findings mean in this context? Are they translatable to HV? Are there any public HV datasets that could be used for validation? To be clear, I am not suggesting generating such a dataset, which would be beyond the scope of this manuscript given the rarity of these tumors.

We thank the reviewer for these questions and agree that this is an important point to clarify: TCGA-BLCA does include HVs, while Chen et al's scRNA-seq dataset includes only UCs. We performed the survival analyses using the Cluster 13 signature in TCGA-BLCA, which we feel is representative since it includes UCs and HVs. Separately, we used Chen et al's dataset to support the rarity of Cluster 13 in UCs. The resistance analyses (CADDRES) were based on RNAseq and drug susceptibility data from the CCLE database, which includes cancer cell lines of all varieties, not just bladder cancers.

While each of these assays on its own is imperfect since a bladder cancer HV-only dataset does not exist, we feel that the number of different databases and variety of analyses included helps us better characterize the nature of the Cluster 13 signature.

2. No across sample harmonization was used. Without this, cluster 13 is the only shared feature that is picked up on analysis. Would this change with harmonization which aims to remove sample specific batch effects? Are other stromal and immune cells shared across samples despite no harmonization? This would potentially justify not harmonizing the tumor cells.

We thank the reviewer for pointing this out. We analyzed our dataset with and without integration; in both cases, Cluster 13 can be identified as a unique population. We ultimately chose to use a non-integrated dataset for our analysis since our stromal and immune cells clustered together across different samples. We included the integrated UMAP with Cluster 13 cells highlighted in as supplemental figure 4 and added a line in our text for clarity:

(Lines 286-287): "This cluster was present in our full dataset with and without integration (Fig. S4)."

3. The authors used InferCNV to confirm tumor cell contents. However, this result could be expanded further upon. Are there any copy number profile differences/similarities between UC vs HV/cluster 13? Could these explain biological differences between the two? Is there a possibility of copy number alteration induced gene dosage effect for genes of interest?

These are great questions. We did not expand on this in our original text because copy number profile appears to vary more among individual tumors than between tumor types. We do not discern any patterns that would distinguish HV vs UC vs Cluster 13. We added a line of text to address this point:

(Lines 280-281): "Of note, we did not discern any patterns in copy number variation between HVs vs UCs."

4. Please add details on the methods for the CaDRReS results. Was this performed on average pseudobulk expression or using the new single-cell version of the package? The citation refers to the older package. Was the pre-existing CCLE model used? If new model training was conducted, please list details of the training parameters used.

We thank the reviewer for this suggestion. In the revised manuscript, we re-trained the CaDRReS-Sc model (the single-cell version) using the GDSC2 database and the CCLE transcriptomic data. Detailed method description (including the model training parameters) has been added to the manuscript:

(Lines 145-153): "CaDRReS-Sc drug response prediction"

Using the tumor cells from each patient (pure UC and variant) and the subsets of Cluster 13 separated by each histological variants (Cluster13_VAR01, Cluster13_VAR03, Cluster13_VAR05, Cluster13_VAR06 and Cluster13_VAR07), we implemented the CaDRReS-Sc method and used the CCLE cell line transcriptomic data (N = 941) and the updated GDSC2 database to train a model to predict the drug response for each cell cluster. A total of 297 drugs were used in the model training. “Cadrres-wo-sample-bias-weight” was selected for the objective function, which is the CaDRReS-Sc model option. 500,000 steps of training were run until the mean squared error (MSE) remained unchanged in 5,000 steps. Then this model was used to infer the drug response (percentage of efficacy) for each tumor cluster in our dataset.”

We also updated the manuscript to reflect new results – we performed a side-by-side analysis of Cluster 13 and parent tumor cells in the 5 HVs with large Cluster 13 contributions, and we observed that the Cluster 13 cells were more resistant to conventional bladder cancer agents than their respective parent tumor cells.

(Lines 342-352): “We next evaluated the susceptibility of Cluster 13 cells to chemotherapy and targeted agents *in silico*. By training a drug response model using the Cancer Drug Response prediction using a Recommender System (CaDRReS) based on the Cancer Cell Line Encyclopedia (CCLE) database and Genomics of Drug Sensitivity in Cancer (GDSC2) database, the estimated efficiency (percentage of tumor cells killed) for drugs from the GDSC2 database was inferred for each tumor cluster in our scRNA-seq dataset (Fig. S8).²⁶ We performed a side-by-side analysis of Cluster 13 and parent tumor cells within VAR01, VAR03, VAR05, VAR06, and VAR07: in 4 of 5 cases, the Cluster 13 subset was more resistant to conventional bladder cancer agents such as cisplatin, gemcitabine, methotrexate, vinblastine, doxorubicin, and mitomycin C compared to their respective parent tumors (Fig. 3C). Consistent with these adverse features, tumors in TCGA-BLCA that harbor higher Cluster 13 signature scores had worse overall survival and disease-specific survival (Fig. 3D).”

Minor:

1. **Pseudotime trajectory callouts should refer to Fig. 2.**

Thank you. This has been corrected.

2. **From Fig. 1B, it is hard to tell if all HVs were represented in cluster 13. Could the authors please revise the color scheme/present this as a table? Based on subsequent results, it seems that certain samples are not represented in cluster 13. Does this correlate with their primary/secondary histology or any other features?**

The color scheme in Fig 1B has been modified for clarity. We also included a column in Fig. 1C with % contribution to Cluster 13. Regarding why certain HV samples were not represented in Cluster 13, there are two possibilities: 1) these tumors did not contain Cluster 13 cells or 2) Cluster 13 cells were not adequately captured either due to discrepancies in tissue processing or inefficiencies in sequencing and quality control. Importantly, our IHC data suggest that most HV tumors harbor cells bear Cluster 13 markers (i.e. CA125) regardless of histology.

Reviewer #3 (Remarks to the Author):

This manuscript entitled “Histologic variants in bladder cancer share aggressive molecular features including a CA125+ cell state and targetable TM4SF1 expression” by Yang et al. studies the tumor biology of histologic variant (HV) subtypes of bladder cancer by performing single-cell RNA sequencing on 9 HV tumors and 3 common bladder cancers. Based on the single-cell RNA-seq data, the authors discovered a CA125+ cell cluster, which is specific to HVs, is responsible for the aggressive biology of HVs. Lastly, the authors demonstrated the utility of CAR-T cells engineered

against TM4SF1 in treating HVs. After carefully evaluating the significance of this work, the conclusion drawn from the results, and the methodology, I suggest that this manuscript is not suitable for publication in Nature Communications. Please see my detailed comments below.

Major points:

1. The authors claimed that cluster13 is mainly contributed by HVs. This conclusion is not very convincing because: 1) the 3 bladder cancer samples contained much fewer cells than HVs (Figure S2C); 2) only 3 bladder cancer samples were used and the UC03 also contained cluster13 cells.

We thank the reviewer for these comments. We acknowledge the limited sample size and cell capture rate of the UC tumors within our dataset. To account for this, we supplemented our analyses with a previously published scRNA-seq dataset of 8 UC tumors comprising >6000 tumor cells (Chen et al, Nat Comms 2020) (Fig. S6). We did not detect a Cluster 13 signature or individual Cluster 13 markers (e.g. MUC16, WISP2, KRT24) within this dataset. Furthermore, our IHC results show that CA125 expression is much more prevalent in HV tumors than UC tumors (13/14 HVs vs 2/20 UCs, $p < 0.0001$, Fisher's exact test). We feel these results support that Cluster 13 is enriched in HVs vs UCs.

2. Line 547, the authors concluded that HVs harbored a special cell cluster expressing TM4SF1, which can be targeted by CAR-T cells. I find this conclusion also unconvincing. Among the 9 HVs subjected to scRNA-seq, cluster 13 cells were present in VAR01, VAR03, VAR05, VAR06, and VAR07 (Figure 1), whereas TM4SF1 was highly expressed in VAR01, VAR04, CAR05, CAR06, and VAR08 (Figure5B). It is unclear whether cluster 13 cells express TM4SF1 or if these are two distinct cell groups.

We thank the reviewer for raising this point as it highlights the need for us to clarify that these are two unrelated and separate conclusions: 1) HVs harbor a Cluster 13 cell state; 2) HVs are also enriched in TM4SF1 expression that is targetable in bladder cancers. While we show in Fig. 5B that Cluster 13 cells express TM4SF1, we do not find that TM4SF1 expression is specific to Cluster 13. We mention this to indicate that a TM4SF1-CAR T cell therapy might also be effective against Cluster 13 cells.

To avoid any potential confusion, we have changed the sentence structure in our concluding paragraph:

(Lines 542-545): "In conclusion, our study demonstrates several new insights regarding HV subtypes in bladder cancer: 1) HVs harbor a clinically significant CA125+ cell subpopulation; 2) HVs are also enriched in expression of a surface antigen that is targetable using CAR T cells; 3) HVs share transcriptional features with histologically similar non-urothelial cancers."

3. The authors claimed cluster 13 cells were more resistant to chemotherapy drugs based on transcriptomic analysis using an online system. It would be more convincing if the authors could isolate these cells based on the cell surface markers, (e.g., CA125) from clinical samples and test their drug responses.

We agree with this comment and have added this as a limitation:

(Lines 536-538): "We also acknowledge that conclusions drawn from *in silico* assays (e.g. chemotherapy resistance) will need to be biologically validated *in vitro* or *in vivo*."

Isolating, culturing, and performing drug sensitivity assays on this cell population (in addition to other biological assays) is the natural next step for this project, and work on this is a future direction.

4. While I understand that the authors put a lot of effort into analyzing the scRNA-seq data of VAR09

and VAR08, Figure 4 and the corresponding text (lines 372-424) were abrupt in this paper. VAR08 and VAR09 did not contain cluster 13 cells, which are the main focus of this manuscript. Additionally, drawing conclusions from a single sample of a particular histologic subtype may not be representative, especially given the disease's heterogeneity. I suggest the authors delete this part from the manuscript.

We appreciate the reviewer's suggestion regarding this section. We feel these findings are important to be included since scRNA-seq data gives us the opportunity to explore the concept of transcriptional mimicry among different cancer types, i.e. to what extent do cancers that look similar behave similarly? These observations can serve to spur future research. We do agree with the reviewer and acknowledge that broad conclusions cannot be drawn from single samples. We have included this in our discussion:

(Lines 538-540): Finally, we acknowledge that the specific patterns of transcriptional mimicry between HV tumors and histologically similar non-urothelial cancers are limited to individual samples used in our study.

5. For CAR-T studies, T cells transfected with empty vector-lentivirus are suggested to be used as control cells instead of untransfected T cells. In Figure 6B, CAR2 cannot kill UMUC3 cells, which contradicts the results shown in Figure S14B. We thank the reviewer for this suggestion. Therefore we performed additional experiments here and an empty vector T cell condition was added (Fig. 6C); the results do not change our conclusions. Regarding CAR2, we did find it to be much less effective than CAR1, which is why CAR1 was chosen for the in vivo experiments. The discrepancy between the two CAR2 in vitro experiments was the E:T ratio (1:1 vs 2:1), which we have now clarified in the figure (Fig. 6B-C). To eliminate confusion, the CAR-T / UMUC3 TM4SF1 knockout experiment was moved to the main figure.

6. It seems that TM4SF1-CAR-T cells only slightly suppressed the growth of UMUC3 cells in vitro (Figure 6B, Figure S14C), but completely killed the tumors in vivo, which is very unexpected.

We agree that the differences in CAR construct (CAR1 vs CAR2) was not made clear in our original figure. We have updated the figure panels with more labels to indicate when CAR1 or CAR2 was used. To address the reviewer's specific comment, CAR1 was able to kill UMUC3 cells in vitro (Fig. 6B), which translates to high killing efficiency in vivo. We also tested CAR1 in vivo against a second cell line (253JBV), which we have now also included in this figure. The text has been modified as follows:

(Lines 462-472): "Finally, we tested CAR1 against xenografts derived from the UMUC3 and 253JBV cell lines (Fig. S18), which were selected for their high TM4SF1 expression and absent NECTIN4 expression. We found that CAR1 exhibited potent anti-tumor activity against these tumors *in vivo* (Fig. 6D). Whereas all untreated control UMUC3 mice died by day 37, all TM4SF1-CAR1-treated UMUC3 mice had a complete and durable response through day 100 (Fig. 6D-E). In the 253JBV cohort, all mice in the untreated control group died by day 107; of the mice that received TM4SF1-CAR1 treatment, three of five (60%) survived through day 160 before reaching tumor size endpoint while two (40%) remained tumor-free through day 199 (Fig. 6E). Importantly, mice treated with TM4SF1-CAR1 cells had stable weights (Fig. S19) and no overt pulmonary toxicity. Taken together, these data demonstrate that TM4SF1 may be a new therapeutic target for HV bladder cancers, including tumors lacking *NECTIN4* expression, and can be successfully targeted using CAR T cell therapy."

Minor points:

- 1. The sample number of HVs in the analysis of serum CA125 levels is too small.** We have increased our sample size and split the UC samples into several categories.
- 2. A picture of mice bearing tumors or resected tumors should be provided in Figure 6, and the number of mice used in each group should be indicated.** We have included the number of mice in each group.

3. Scale bars were missing in Figure 5D and Figure S1, and were not annotated in the figure legends for Figure 5C. We apologize for the omission. These figures have been updated with scale bars.

4. The authors removed the correlation plots of TM4SF1 and EMP1, EZR, CLDN4, and KRT19 to Figure S12, but the figure legend remained in Figure 5. We thank the reviewer for this comment. The figure legend has been updated.

5. Figure S14, TM4SF1 knockout efficiency in UMUC3 cells should be indicated. TM4SF1 knockout efficiency was quantified using flow cytometry. This has been added to the figure.

6. Line 225, why can a biotinylated c-myc antibody be used to sort transfected cells? We thank the reviewer for raising this point of clarification. The CAR construct itself has an extracellular Myc tag, which is why this method can be used to not only verify surface expression of the CAR (using an anti-Myc-647 antibody) but also for isolation. We have added this to the methods for clarity:

(Lines 186-188): “The VH and VL sequences were cloned in two configurations using the Gibson Assembly protocol (Twist) into a CAR backbone containing a myc tag, IgG4 spacers, CD8 hinge and transmembrane domain, 4-1BB costimulatory domain, CD3 ζ chain, and EGFP.

We used the anti-myc biotinylated antibody (Miltenyi, 130-124-844) plus the anti-biotin magnetic microbeads (Miltenyi, 130-090-485) to magnetically separate the CAR T cells on MS Separation Columns (Miltenyi, 130-040-201), if the lentiviral transduction efficiency of the T cells was <30%. This type of separation is different than FACS (fluorescent-activated cell sorting) because there are no fluorophores involved. We chose to use this method instead of FACS because: 1) we were occasionally having issues with contamination after running our primary T cells through the BD Fusion and this magnetic separation method can be done in the biosafety cabinet in a completely sterile environment, and we did not want to risk having a contamination; and 2) the magnetic separation is more gentle on the primary T cells because it is done on a column and gently eluted (as opposed to cells being pushed through the Fusion sorter), and so the viability and expansion of the T cells afterwards was generally better.

7. Fonts should be unified in figures, and font size should be increased in Figure S11B. We thank the reviewer for pointing this out. All fonts have been changed to Helvetica. Fig. S11B has been updated.

Reviewer #3 (Remarks on code availability):

Single-cell RNA-seq files and all code used in the present study are not available in this manuscript. All raw data will be uploaded to the Gene Expression Omnibus (GEO), and code will be made available in a Github repository upon publication.

RESPONSE TO REVIEWERS' COMMENTS

My overall assessment is that the authors have made a good effort to address the points raised by original reviewer number 3:

In my opinion major points 1-3 were sufficiently addressed by the authors. For instance, the authors included an additional data set of >6000 UC cells to strengthen the point that Cluster 13 cells are mainly specific for the HV subset.

I agree with the assessment of reviewer 3 regarding the data presented in Figure 4 (major point 4). To me this part also seems to be disconnected from the rest of the story. I would suggest to rather put this data to the supplement and to also deemphasize this part in the text.

We thank the reviewer for the thoughtful acknowledgement of our revisions and for the suggestion to put these data into the supplement. While we believe that these results would be of interest to our colleagues studying bladder cancer, we agree that this section seems out of place in the overall flow of our manuscript. Therefore we have dramatically condensed and de-emphasized these data as requested and now mention in the text the transcriptomic similarities between HVs and non-urothelial cells in the first section as follows:

Line 201: "Although there were few apparent transcriptional similarities among the three micropapillary tumors or between the two nested tumors, we did find an enrichment of genes related to plasma cell maturation²⁴⁻³⁵ and small cell lung cancer³⁶⁻³⁸ within the top differentially expressed genes (DEGs) in VAR08c (derived from plasmacytoid HV) and VAR09c (derived from small cell HV), respectively (Supplementary Fig. 4A-D)."

The section and main figure have otherwise been removed.

Regarding the CAR T cell related part:

The points raised by original reviewer number 3 were sufficiently addressed in my opinion.

1. In Figure 6E right panel the authors write in the text that some mice survived until day 199 and others had to be taken out at day 160. However, the survival curve ends at day 150. To reflect the entire picture the survival curve until day 160 should be displayed.

We thank the reviewer for this comment. We have revised the figure as requested and the survival curve has been extended to display day 160.

2. How did the 253JBV tumors escape the CAR T therapy? Did they lose target expression?

We thank the reviewer for asking this important question and it is one we are currently working to address to characterize the 253JBV tumors that escaped therapy. We speculate that downregulation of the protein is one mechanism that may be happening in these cells. Unfortunately, the antibody that we initially used for our studies has been discontinued recently so it would take additional time to validate a new antibody. While we are excited to learn the answer to this question, we feel it would not significantly

impact the major conclusions of our study. Given the time it will take to generate these results, we aim to use these results in a future study.

3. Do the cell lines used for the CAR T studies represent HV tumors to justify the following conclusion: “Taken together, these data demonstrate that TM4SF1 may be a new therapeutic target for HV bladder cancers, including tumors lacking NECTIN4 expression, and can be successfully targeted using CAR T cell therapy”.

We thank the reviewer for raising this point, and we acknowledge that our original statement needs clarification. We have changed the wording as follows:

Line 539: “Taken together, these data demonstrate that TM4SF1 may be a new therapeutic target for TM4SF1-expressing bladder cancers, including tumors with variant histology and those lacking NECTIN4 expression, and can be successfully targeted using CAR T cell therapy in a mouse xenograft model.”

4. What is the expression pattern of TM4SF1 on healthy tissue? The authors did not see any signs of toxicity with these CAR T cells in mice (cross reactivity of the original antibody with murine tissue?) but it would be good to at least comment on potential on target off tumor toxicities of TM4SF1 targeting CAR T cells.

We agree this is an important consideration for a preclinical study. We have added the following data and statement to acknowledge this:

Line 542: “Human toxicity remains a concern, given the expression of TM4SF1 in other human tissues (Supplementary Fig. 20), so further studies are needed to characterize and mitigate these potential effects.”